# Nano-Sheet-like Morphology of Nitrogen-Doped Graphene-Oxide-Grafted Manganese Oxide and Polypyrrole Composite for Chemical Warfare Agent Simulant Detection

**DOI:** 10.3390/nano12172965

**Published:** 2022-08-27

**Authors:** Sanjeeb Lama, Bong-Gyu Bae, Sivalingam Ramesh, Young-Jun Lee, Namjin Kim, Joo-Hyung Kim

**Affiliations:** 1Laboratory of Intelligent Devices and Thermal Control, Department of Mechanical Engineering, Inha University, Incheon 22212, Korea; 2Department of Mechanical, Robotics and Energy Engineering, Dongguk University, Seoul 04620, Korea; 3Department of Mechanical Engineering, Jeju National University, Jeju 63243, Korea

**Keywords:** chemical warfare agents (CWAs), quartz crystal microbalance (QCM), surface acoustic wave (SAW), dimethyl methyl phosphonate (DMMP), volatile compounds (VOCs)

## Abstract

Chemical warfare agents (CWAs) have inflicted monumental damage to human lives from World War I to modern warfare in the form of armed conflict, terrorist attacks, and civil wars. Is it possible to detect the CWAs early and prevent the loss of human lives? To answer this research question, we synthesized hybrid composite materials to sense CWAs using hydrothermal and thermal reduction processes. The synthesized hybrid composite materials were evaluated with quartz crystal microbalance (QCM) and surface acoustic wave (SAW) sensors as detectors. The main findings from this study are: (1) For a low dimethyl methyl phosphonate (DMMP) concentration of 25 ppm, manganese dioxide nitrogen-doped graphene oxide (NGO@MnO_2_) and NGO@MnO_2_/Polypyrrole (PPy) showed the sensitivities of 7 and 51 Hz for the QCM sensor and 146 and 98 Hz for the SAW sensor. (2) NGO@MnO_2_ and NGO@MnO_2_/PPy showed sensitivities of more than 50-fold in the QCM sensor and 100-fold in the SAW sensor between DMMP and potential interferences. (3) NGO@MnO_2_ and NGO@MnO_2_/PPy showed coefficients of determination (R^2^) of 0.992 and 0.975 for the QCM sensor and 0.979 and 0.989 for the SAW sensor. (4) NGO@MnO_2_ and NGO@MnO_2_/PPy showed repeatability of 7.00 ± 0.55 and 47.29 ± 2.69 Hz in the QCM sensor and 656.37 ± 73.96 and 665.83 ± 77.50 Hz in the SAW sensor. Based on these unique findings, we propose NGO@MnO_2_ and NGO@MnO_2_/PPy as potential candidate materials that could be used to detect CWAs.

## 1. Introduction

Chemical warfare agents (CWAs) are considered extremely lethal weapons that have the potential to cause mass destruction [1]. The CWA attacks on the Tokyo subway (1995) [2] and the Syrian civil war (2013) [3] demonstrated to the world the ramifications of the misuse of CWAs. Based on this, we understand that the early detection of CWAs is crucial for preventing the loss of human lives. Appendix A shows the chemical structure of G-series nerve agents, such as sarin (GB series), soman (GD series), Tabun (GA series), and simulant DMMP. For many years, detection technologies have been suggested and developed to detect CWAs. Some of these include gas chromatography, Raman spectroscopy, Fourier-Transform Infrared Spectroscopy, atomic emission detection, and ion mobility spectroscopy [4,5]. This equipment is used for the detection of both CWAs and their simulants, and it is accurate, precise, and reliable.

However, this equipment has crucial problems when used in field monitoring of CWAs. For practical purposes, it is important to detect CWAs in the field. For the field monitoring and detection of CWAs, devices with less power consumption, high accuracy, portability, reliability, inexpensive manufacturing and maintenance costs, and stability are distinctly favored. In order to fulfill all these requirements for field monitoring of CWAs, acoustic wave sensors, such as SAW and QCM sensors, have been employed in the present [6,7]. Appendix A represents the list of acronyms and its definition.

QCM and SAW sensors are used in various fields because of their feasibility of usage for gaseous and liquid measurements. A QCM sensor operating as a bulk acoustic wave sensor utilizes the electromechanical resonant behavior that is highly sensitive to the change in mechanical properties, such as the mass, damping, and stiffness. The sensed quantities are then transduced into electrical measures for the detection of change in the mechanical properties. QCM has been employed in the detection of infectious diseases, VOCs, environmental pollutants or nanoparticles, pathogens or bacteria, humidity, cells, and biomolecules as well as in the assessment of food quality [8].

Hewa et al. proposed to use QCM sensors for sensitive and rapid detection of the influenza A and B virus [9]. A QCM sensor that uses calixarenes as sensing thin films was developed to detect different VOCs, such as alcohols, ketones, aromatics, and chlororganics [10]. A QCM sensor along with an integrated microheater was used to detect particulate matter with adverse effects on human health and the environment [11]. Buchatip et al. proposed the detection of pathogenic bacteria, *Vibrio harveyi*, using a QCM sensor [12].

This bacterium is well known to cause morbidity and fatality in shrimp culture. A novel sensing material named silica SBA-15 was studied by Zhu et al. [13] to be used along with monodisperse hexagonal lamelliform in QCM sensors to detect humidity. This detection of humidity is necessary in drug storage, industrial production, agriculture, and other medical applications.

Escuderos et al. used a QCM sensor to detect the high level of olive oil sensory significance, which can help prevent defects produced by olive biogenic pathways, olive fruit over-ripening, bacteria attacks caused by inappropriate harvesting, and inadequate manipulation of olive paste [14]. Despite these numerous advantages, QCM sensors have several limitations, including the low resonant frequency obscuring its sensing performance [15] and the electrode structure influencing the sensitivity profile and displacement in conventional sensors that may degrade the sensing performance [16].

On the other hand, SAW sensors are sensitive to perturbations of the analyte adsorbed onto the delay line along with the changes in their phase velocity and amplitude, which is further transduced into the electrical parameters [17]. SAW sensors possess several advantageous features, such as small size, high reliability, low cost, mechanical stability, good reproducibility, and energy efficiency, which are all made possible primarily due to the usage of concepts and technical procedures from micro-electromechanical system (MEMS) technology [18].

Since its introduction, SAW sensors have been used in the detection of chemical vapors and gases, biomolecules, bacteria, cells, carbon dioxide, humidity, electric power temperature, pressure, and strain measurements. Deng et al. studied hierarchical Co_3_O_4_ nanorods exhibiting speedy response and recovery with high magnitude in the detection of ammonia gas [19]. The use of SAW sensor in the detection of carbon dioxide in high temperature environments was proposed [20].

Wessa et al. [21] exploited a SAW sensor in the immunosensing of photo-bonded proteins, which showed fast interactions with the antibody or antigen. The determination of the enzyme–substrate complex during the catalytic reaction formed on the glucose oxidase was studied by Inoue et al. [22]. A SAW sensor was also used to examine the sensing performance in relative humidity (R.H.) conditions ranging from 10–80% [23]. Heider et al. proposed the use of a SAW sensor to wirelessly sense the temperature on the crankshaft of naval engines [24].

Regarding liquid measurements, the SAW sensor has been used for the detection of chemical and biochemical substances [25]. Although it exhibits several useful characteristics, the SAW sensor has disadvantages, such as a higher frequency resonator resulting in low accuracy, less ability in solving minute changes in the measurand, higher aging rates, and the requirement of more power [26]. In the case of liquid measurements, the main disadvantage is the reflection of longitudinal waves in the SAW sensor [27].

In the pursuit of discovering highly sensitive functionalized material for the detection of different vapors, carbon nanotubes [28], metal oxides [29], conducting polymers [6,30], and metal organic frameworks [31] have been developed thus far. Manganese dioxide is one of the materials used in the detection of CWAs that has applications in gas sensors [32], energy storage, water treatment, batteries, supercapacitors, and decomposition. PPy is a conducting polymer that has affinity towards gas sensors and that has been used with its hybrid composites to detect DMMP (*M.W*. = 124.08 g/mol), methanol, ammonia, hydrogen sulfide, nitrogen dioxide, and carbon dioxide [33,34].

It was reported that the incorporation of ZnO with manganese dioxide effectively absorbed DMMP vapor, producing significant data for CWA simulant detection [35]. Another approach for CWA simulant detection is the use of hybrid composite materials containing PPy that showed excellent sensor responses [36]. Another study [37] on graphene oxide demonstrated that it generated remarkable sensitivity, repeatability, and stability for CWA simulant detection. Based on the Tanimoto coefficient and Euclidean distance [38], DMMP is one of the preferred simulants of the GB nerve agents. Appendix A illustrates the chemical structure of the synthesized hybrid composite materials.

In this study, we present the characterization of hybrid composite materials and study their morphology using Fourier-Transform Infrared Spectroscopy (FTIR), X-ray Diffraction (XRD), X-ray Photoelectron Spectroscopy (XPS), Scanning Electron Microscopy (SEM), and Transmission Electron Microscopy (TEM). NGO@MnO_2_ and NGO@MnO_2_/PPy were deposited in the QCM and SAW sensors to test their frequency shifts, selectivity, polar plots, linearity, repeatability, and response/recovery times. The sensing performance of the fabricated composite materials was analyzed and compared for both the sensors. The effect of R.H. on the sensing performance was also investigated. Note that the used acronyms in this paper are listed in Appendix A.

## 2. Materials and Methods

### 2.1. Materials

Graphite flakes, pyrrole, hydrogen peroxide (H_2_O_2_), sodium nitrate (NaNO_3_), manganese (III) acetate dihydrate, sulfuric acid (H_2_SO_4_,), potassium hydroxide (KOH), urea, N-methyl-2-pyrrolidone, ferric chloride (FeCl_3_), potassium permanganate (KMnO_4_), ammonium hydroxide (NH_4_OH), polytetrafluorethylene (PTFE), and DMMP were acquired from Sigma Aldrich, Seoul, Korea. We also used 95% Ethanol (Merck, Darmstadt, Germany), 95% n-hexane (Avantor, Radnor, PA, USA), 99.5% Toluene (Duksan, Incheon, Korea), 99.5% Isopropyl alcohol (IPA) and 99.80% methanol (Daejung, Siheung, Korea) in the fabrication.

### 2.2. Fabrication of Composite Materials

#### 2.2.1. Synthesis of NGO

An Improved Hummers method was used to synthesize GO materials. This method is reported in the literature [39]. The calculated amount of graphite flakes (30 g) and NaNO_3_ (7 g) were placed together with H_2_SO_4_ (300 mL) at a temperature of 0 °C in an ice bath. Next, 6 g of potassium permanganate (KMnO_4_) was added slowly to the mix kept in the bath, and the temperature of the bath solution was increased to 50 °C with 12 h of continuous stirring. At the end of the process, the color of the resultant product turned greenish brown. Subsequently, filtration of the resultant product was completed, and the product was rinsed with water and ethanol several times. Hence, the obtained GO was heated for 24 h at 50 °C in the vacuum oven.

In order to prepare NGO, the prepared GO (3 g) was dissolved into water (300 mL) and sonicated for 4 h to allow for uniform dispersion of the GO. After proper dispersion, the solution was filtered using a Millipore filter. The filtered solution was then dried for 12 h at 95 °C in an oven. The filtered GO solution was mixed with ammonia and urea (10 mL) together with continuous stirring for 12 h at 95 °C to acquire a homogenous solution. The solution obtained out of this process thus far was dried in an oven for 12 h at 180 °C and then purified further using an ethanol solvent. As the last step, the resultant product was calcined for 12 h at 200 °C.

#### 2.2.2. NGO@MnO_2_ Composites

In order to prepare the NGO@MnO_2_ composites, a calculated amount 0.6 g of as-prepared NGO (0.01 M), manganese acetate and KMnO_4_ (0.01 M) were dissolved in double-distilled (DD) water in a glass vessel of 500 mL. The ammonia solution (25 mL) was introduced into the solution in the glass vessel and stirred continuously for 12 h at 95 °C. The solution obtained after stirring was then transferred to a Teflon-lined autoclave (300 mL) and roasted for 12 h at 180 °C. After the roasting was completed, the resulting product was purified using Millipore water and dried for 12 h at 95 °C. At the end of the drying process, the collected composite sample was calcinated for 10 h at 450 °C in a vacuum furnace.

#### 2.2.3. NGO@MnO_2_/PPy Composite Synthesis

The hybrid composite was prepared using the following procedure. First, the as-prepared NGO (0.6 g), manganese acetate (0.01 M), KMnO_4_ (0.01 M), pyrrole (6 g), and FeCl_3_ (1.5 g) and H_2_O_2_ (10 mL) were blended into 100 mL of DD water in a glass vessel. The solution obtained after blending was continuously stirred for 4 h at room temperature to acquire a homogenous dispersion. Subsequent to this stirring, the solution was allowed to undergo in situ oxidative polymerization for 2 h at 90 °C. After the polymerization was finished, the solution was transferred to a Teflon-lined autoclave and heated there for 10 h at 450 °C. After heating, the resulting product was collected and used in our experiments without further processing. Figure 1 shows the schematic diagram illustrating the fabrication process of NGO@MnO_2_/PPy.

### 2.3. Target Vapor Preparation

Figure 2a illustrates the bubbler flask used in the presented study. The 500 mL bubbler flasks were cleaned using ethanol for several times and dried in an oven for one hour at 60 °C. After the chemicals were received, they were poured into the pre-cleaned bubbler flask with the help of a pre-cleaned glass funnel. Then, two steel tubes attached with Teflon cork were inserted into the inlet and outlet of the bubbler. The steel tubes allowed nitrogen and air to flow through the inlet to outlet of the bubbler. The Teflon cork made the bubbler system airtight to prevent the leakage of generated target vapor. After the bubbler flask was used in the experiments, it was stored in a Chemsafe systems Storage Cabinet (Samillab, AL-D 1002).

### 2.4. Sensor Measurement System

The gas response measurement system is divided into three major parts—namely, a vapor-generating bubbler, analyte delivery system, and detection chamber and data acquisition. In our previous works [6,40], we briefly introduced the gas response measurement system. In this presented study, each part will be explained in detail.

#### 2.4.1. Vapor-Generating Bubbler

A two-necked bubbler was used to vaporize the liquid simulant and VOCs for the gas response measurements. Dry N_2_ gas for QCM (dry air for SAW) acting as a carrier gas enters into the base of the target vapor containing bubbler through the left inlet steel tube. Then, the carrier gas is distributed all over the target liquid sample, which takes the analyte vapor upward and finally through the right outlet steel tube. Figure 2b shows the schematic of the vapor generating process in the presented work. The output flow rate of analyte vapor, *F*_s_, can be formulated using a bubbler equation [41]:(1)$$F_{s}=\left(\frac{P_{\mathrm{th}}\times F_{c}}{P_{0}-P_{\mathrm{th}}\;}\right)=\alpha \;F_{c}$$
where *F_c_* is the flow rate of carrier gas in sccm (standard cubic centimeters per minute), *P*_0_ is the outlet pressure in the bubbler headspace above the target liquid, and *P*_th_ is the thermodynamic vapor pressure of the target analyte.

The vapor pickup efficiency (α) is defined as the relative ratio of the output target vapor flow rate by the carrier gas flow rate. These parameters were acquired under the standard conditions of *T* = 298 K and *P* = 760 mmHg (=1 atm) in the presented study. *P*_th_ converts to *P*_s_ (saturated vapor pressure) when the carrier gas is totally saturated with the target analyte vapor. *P*_s_ of the target vapor can be calculated from the Antoine Equations (2) and (3) expressed as below [42]:(2)$$\ln\;P_{s}\;\left(A-\frac{B}{C+T}\right)$$
(3)$$\log\;P_{s}\;\left(A-\frac{B}{C+T}\right)$$
where *T* is the temperature of the bubbler controlled by a digital water bath (LabTech, LWB-106D). The value of *P*_s_ is greatly dependent on the temperature of target liquid containing bubbler controlled by the digital water bath. The main reason is that the output flow rate was calculated under the fixed bubbler temperature and constant carrier gas flow rate. *A*, *B*, and *C* are the coefficients used in the Antoine equation for various vapors, which is given in Appendix A.

#### 2.4.2. Analyte Delivery System

The generated target vapor was transported from the bubbler headspace to the detection chamber. The gas feeding system consisted of nitrogen cylinders connected to the mass flow controllers (MFCs) (KOFLOG, 3660), one-way valves, a vapor bubbler system, steel pipes, and one-touch connectors as shown in Appendix A. The carrier gas flow was controlled by setting the MFC 2 flow rate. The generated target vapor was mixed with the dilution gas flow, *F*_d_ (2000 sccm in QCM and 1000 sccm in the SAW sensor), by setting the MFC 1. The resultant target vapor concentration, *C*_vapor_, in ppm (parts per million) was enumerated from the predetermined output flow rate and the dilution ratio [43]:(4)$$C_{\mathrm{vapor}}\;(\mathrm{ppm})=\left(\frac{\;F_{s}\times 10^{6}}{F_{d}+F_{c}+F_{s}}\right)=\left(\frac{\alpha \;\times 10^{6}}{(F_{d}\;/\;F_{c})+1+\;\alpha }\right)$$

In the present study, four VOCs (ethanol, methanol, n-hexane, and Toluene) and water were detected using a fixed carrier flow rate (200 sccm in QCM and 100 sccm in SAW) and a bubbler temperature of 25 °C. As the objective of this study was to detect DMMP, the carrier flow rate was varied from 48 to 320 sccm in QCM and 24 to 160 sccm in the SAW sensor with a fixed bubbler temperature of 25 °C. The dilution flow rate was kept constant most of the time (2000 sccm in QCM and 1000 sccm in the SAW sensor). In both the QCM and SAW sensors, the target vapor was fed to the detection chamber alternately for 5 min. The dilution flow was continuously fed to the detection chamber at the fixed flow rate for enough time to reach the sensor steady baseline.

#### 2.4.3. Detection Chamber and Data Acquisition

In the presented work, a QCM controller system (QCM200, Stanford Research System, Sunnyvale, CA, USA) was used to measure the sensing performance of the hybrid composites (See Appendix A). Regarding the detection chamber or flow cell, after the installation of axial flow cell adapter, a stagnation point flow cell with a small volume was created. The volume of flow chamber was around 0.15 mL. Appendix A shows the QCM flow cell used in the presented study. The flow cell is built using the chemical resistant Kynar.

It consists of two ports—inlet and outlet—with a 0.040 inch through-hole equipped with barbed hose adapters with 0.062 inch tubing. The flow cell utilizes the axial flow for the adsorption and desorption processes. In the axial flow cell, the gas flows inwards perpendicularly and radially outward from the stagnation point to the exit at the edge of the cell in a volume of 150 µL. The stagnation point is positioned at the center of the QCM crystal electrode where there is zero hydrodynamic flow or absence of surface shear forces [44]. Appendix A illustrates the flow pattern in an axial flow cell.

A QCM200 Digital controller displays the measured resonant frequency of the QCM crystal at the interval of 1 s, which is transferred to a personal computer. The data acquisition was done by software called SRS QCM200 (see Appendix A).

The SAW sensor setup used in the presented study was developed in the laboratory clean booth. It consists of a detection chamber, test board, vector network analyzer (VNA-MS46122A, Anritsu, Richardson, TX, USA), and SMA connectors (see Appendix A). Regarding the detection chamber, it has a volume of 420 mL. Appendix A show the top and front views of the detection chamber without and with the top cover, respectively. The SAW sensor detection chamber applies a similar principle with the QCM flow cell.

The gas flows perpendicular towards the test board, which holds the SAW sensor, and exits radially outward from the edge of the chamber. Appendix A shows an illustration of the gas flow in the SAW detection chamber. VNA was used to transfer the raw sensor signal to the personal computer. The data acquisition was done at intervals of 10 s with the help of software called Shockline. Appendix A shows the SAW sensor signal monitoring program.

Gas response measurements were performed by positioning the hybrid-composite-coated QCM and SAW sensors inside the flow cell or detection chamber and blowing the diluted target vapor over them while concurrently monitoring the frequency changes of the sensors. All the experiments were conducted inside the clean booth in which the temperature and humidity was regulated with the air-conditioned environment maintained at 20 °C (LG Whisen-LPNW1451VJ, LG, Seoul, Korea).

### 2.5. QCM and SAW Sensor

The QCM200 system consists of a built-in frequency counter and resistance meter. It includes a digital controller, crystal holder, crystal oscillator, flow cell adapter, and quartz crystals. QCM crystals are AT-cut quartz crystals embedded between two conducting chrome/gold electrodes operating at 5 MHz with a diameter of 1 inch. Appendix A shows the QCM crystal used in the presented study. In 1880, the Curie brothers introduced the piezoelectric effect in piezoelectric crystals for the first time [45].

In 1959, Sauerbrey [46] discovered an expression relating to the measurable changes in the crystal’s resonant frequency (∆*f*) with the mass deposited (∆*m*) in gaseous/vacuum media, i.e., ∆*f* = −*C*_f_ × ∆*m*. The sensitivity factor (*C*_f_) is 56.6 Hz/(µg·cm^−2^) for AT-cut quartz crystal of 5 MHz at the room temperature [44]. The AT-cut crystals are well known for their zero-temperature coefficient at room temperature. Hence, the inbuilt temperature dependence of QCM crystals is insignificant at room temperature (~1–3 Hz/°C). The operating temperature of the QCM system is 0–40 °C. However, all the QCM experiments in the present study were conducted at 20 °C.

In 1885, Lord Rayleigh introduced surface acoustic waves propagated along the plane surface of an elastic solid [47]. R. M. White discovered the direct piezoelectric surface wave transduction by using the interdigital transducers (IDTs) [48]. Appendix A shows a schematic of the 250 MHz SAW sensor used in the presented study. SAW sensors were forged out of a substrate made of ST-cut quartz. The sensor also consists of an aluminum alloy with 1% copper based interdigital transducer (IDT) and a 50 nm Ti adhesive layer deposited using an e-beam evaporator.

The sensitivity of the SAW sensor is influenced by several factors, including the piezoelectric materials, center frequency, wavelength, and SAW velocity. The central frequency (*f*_c_) of the SAW sensor is directly proportional to the SAW velocity (*v*) and inversely proportional to the wavelength (*λ*), i.e., *f*_c_ = *v*/*λ* [49]. The center frequency of the SAW sensor was designed to be 250 MHz, the velocity of SAW on the quartz substrate was 3158 m/s, and the wavelength of the aluminum electrode was 12.632 µm [50].

#### 2.5.1. Deposition of Sensing Material on the QCM Sensor

The synthesized NGO@MnO_2_ and PPy grafted NGO@MnO_2_ were intermixed with IPA in a 10 mg: 1 mL ratio. The solutions were then ultrasonicated for 3 h. After this, 15 µL of this mixture was drop-coated on the QCM surface. The sensing material deposited QCM sensors was then kept in an oven for one hour at 60 °C. Once removed from the oven, the QCM sensors were allowed to cool at room temperature of 20 °C and were used in measurements.

#### 2.5.2. Deposition of Sensing Materials on the SAW Sensor

The sensing material was deposited on the SAW sensor using the following procedure. First, the prepared composite materials were intermixed with IPA in 1 mg:7 mL ratio. Since SAW sensors are highly sensitive to the weight of deposited sensing materials, this ratio was changed from the one previously used for the QCM sensor. The mix was then ultrasonicated for 3 h. After the ultrasonication, 2.5 µL of the solution was drop-coated onto the top surface of the SAW sensor. The SAW sensors deposited with thin sensing films were then heated in an oven at 60 °C for one hour. Once the heating was complete, the obtained SAW sensors were allowed to cool to room temperature at 20 °C.

### 2.6. Characterization Methods

#### 2.6.1. Material Characterization Apparatus

The hybrid composites were characterized using Vertex 80v FT-IR Spectrometers (Bruker, Billerica, USA) that provide excellent sensitivity and stability. The XRD patterns of the hybrid composites were studied using high resolution X-ray diffractometry—X’pert PRO MRD (Philips/Panalytical, Malvern, UK). FE-SEM—S-4300SE (Hitachi, Tokyo, Japan) was used to study the morphological analysis of the fabricated hybrid composite materials. FE-TEM—JEM-2100F (Jeol, Akishima, Japan) was used to characterize the hybrid composites to study its structure, thickness, and diameter. The elemental analysis of the composite was performed using X-ray photoelectron spectroscopy—K-Alpha (Thermo Fisher Scientific, Waltham, MA, USA).

#### 2.6.2. Gas-Response System Characterization Apparatus

The weight of the sensing materials was measured by analytical balance—AS200.R2 PLUS (Radwag, Radom, Poland). The ultrasonication processes were done using the sonicator—POWERSONIC 410 (Hwashin Technology, Seoul, Korea). An oven—LDO-150F (Daihan LabTech, Namyangju, Korea) was used to heat the bubbler, QCM crystal, and SAW sensor at 60 °C. After the QCM and SAW sensors were used, Petri dishes with the sensors were kept in the vacuum desiccator—VL-B (MadeLab, Hannam, Korea). The outlet of the QCM and SAW chambers were connected to the Fumehood—SFH series 1000/2011 (Shinsaeng, Yonggin, Korea). The R.H. test was conducted using a temperature and humidity chamber—TH-ME-100 (Lab companion, Daejeon, Korea). The temperature and humidity inside the clean booth were measured using a temperature humidity meter—Fluke 971 (Fluke, Everett, D.C., USA) and ETP101 (All-sun, Zhangzhou, China).

## 3. Results

### 3.1. FT-IR and XRD Analysis

Appendix A shows the FT-IR spectrum analysis of the hybrid composites. In the FT-IR spectrum, the peak position at 2918–3447 cm^−1^ represents the adsorption bands of N–H, C–H, and O–H, respectively [51,52]. The intense peak at 1630–1715 cm^−1^ corresponds to the C=O stretching and C=C stretching adsorption bands of the carboxylic group. C–N in-plane vibration is observed at the intense peak of 1364–1385 cm^−1^ [51]. It is important to note that these adsorption bands primarily arise from the fabrication of NGO.

In addition, the lower wave number from 623–415 cm^−1^ represents the stretching vibration of the Mn–O–Mn and Mn–O bonds, which, in the present work, indicates the formation of MnO_2_ [52]. Finally, PPy shows the peak position from 3447–415 cm^−1^, which represents C–H stretching, C–H in-plane deformation, C–H out-of-plane vibration, C=C stretching of pyrrole ring, C–N stretching, and N–H stretching [53].

This also indicates the formation of PPy in the composite materials via a hydrothermal process. Hence, the FT-IR spectrum analysis confirmed the presence of MnO_2_, PPy, and NGO in the samples prepared as part of the presented work.

The XRD analysis of the NGO@MnO_2_ composite showed peaks at 12.5°, 25.4°, 27.0°, 36.0°, 37.5°, 38.5°, 40.0°, 42.7°, 47.7°, 50.8°, 63.3°, 65.4°, and 69.8°, which represent the planes 110, 220, 310, 400, 211, 330, 301, 510, 411, 600, 521, 002, and 541, respectively (see Appendix A). The resulting peaks and planes depict the polycrystalline structure as well as the tetragonal phase of MnO_2_ (JCPDS: 44-0141) [54] on the NGO surface. The peak position of 2*θ* = 25.4° corresponds to the heap of NGO sheets.

Similarly, the observed peaks of NGO@MnO_2_/PPy match with the lattice plane confirming the crystalline structure and the tetragonal phase of α-MnO_2_ (JCPDS: 240734 and 44-0141). We note that PPy might have arisen from the XRD peak between 2*θ* = 25.4°and 28.4–33.1° (see Appendix A) [54,55]. In Appendix A, a low intensity peak of NGO was observed between 2*θ* = 25.4° and 20.0–25.0°. From all these observations, we can conclude that the XRD results confirm the presence of NGO and PPy in NGO@MnO_2_/PPy hybrid composite materials.

### 3.2. X-ray Photoelectron Spectroscopy (XPS)

To study the elemental composition of the hybrid composites, we used X-ray photoelectron spectroscopy. The XPS analysis showed the presence of elements, such as Mn, O, C, and N, and this is illustrated in Appendix A. In Appendix A, 2*p*_1/2_ (654 eV) and 2*p*_3/2_ (642 eV) are shown as the characteristic peaks of Mn 2*p*. The energy difference between the 2*p*_1/2_ and 2*p*_3/2_ is 12 eV [55]. In Appendix A, an O 1*s* characteristic peak is shown at 530 eV, which is mainly due to the bonding of metal and oxygen during the synthesis.

In Appendix A, C 1*s* characteristic peaks are shown in 295, 293, and 285 eV corresponding to O–C=O, C–C *sp*^3^, and C–C *sp*^2^, respectively, of the NGO@MnO_2_ composite indicating oxidation at a high degree [56]. In Appendix A, the N 1s characteristic peak is shown at 407 eV, which is primarily due to graphitic N in NGO. This also indicates that nitrogen is successfully infused into the hybrid composites. Finally, Appendix A shows the elemental spectrum of the NGO@MnO_2_ composite, which confirmed the presence of Mn, O, C, and N.

Similarly, for NGO@MnO_2_/PPy, the characteristic peaks of Mn 2*p* were observed at 653 eV (2*p*_1/2_) and 641 eV (2*p*_3/2_), which is illustrated in Appendix A. The energy difference between 2*p*_1/2_ and 2*p*_3/2_ is 12 eV [55]. In Appendix A, the O 1*s* characteristic peak is shown at 530 eV because of the OH group in NGO arising from bonding of metal and oxygen during the fabrication process. In Appendix A, C 1s characteristic peaks were seen at 295 eV (O–C=O), 293 eV (C–C *sp*^3^), 289 eV (C–N), and 285 eV (C–C *sp*^2^) in the NGO@MnO_2_/PPy composites [54,56]. In Appendix A, N 1*s* characteristic peaks were shown at 406 eV (graphitic N) and 400 eV (pyrrolic) in NGO [57]. Finally, Appendix A illustrates the presence of Mn, O, C, and N in the NGO@MnO_2_/PPy composite.

### 3.3. SEM and TEM

SEM and TEM analysis were used to study the morphological structures of the hybrid composite high-resolution SEM images. The energy dispersive X-ray analysis (EDX) profiles of the hybrid composite are illustrated in Figure 3. Figure 3a–f illustrates that the smaller particles are stacked haphazardly and formed a cauliflower-like morphology resulting from an aggregation of minute particles. The primary reason for this agglomeration is the incorporation of NGO onto the MnO_2_ [54].

The size of the fabricated MnO_2_ nanoparticles ranges from ~20 to 100 nm in the composites prepared. The observed size of the grains within the composites was ~20–30 nm. Figure 3g–i illustrates the EDX profile, which confirms the presence of Mn, O, N, and C. We also found that the elemental percentages by weight of Mn, O, N, and C were 85.5%, 9.0%, 1.0%, and 4.5%, respectively. This is convincing evidence for the formation of MnO_2_ and NGO in the synthesis of NGO@MnO_2_.

Similarly, Figure 4a–f shows the high-resolution SEM images of NGO@MnO_2_/PPy. They depict the formation of a cluster of small grain-like structure mainly due to the formation of NGO on the MnO_2_ structure. The superficial side of the composites has a spherical grain-like structure (see Figure 4d,e) primarily due to the infusion of PPy onto the NGO@MnO_2_ to eventually form the NGO@MnO_2_/PPy composite. The inclusion of PPy is expected to improve the surface area [58] and the conductivity that results from π-electron conjugation [59], which is highly preferable for adsorption/desorption kinetics.

Figure 4g–i illustrates the EDX profile of NGO@MnO_2_/PPy depicting the elements composed, such as Mn, O, N, C, and Fe. The main reason for the appearance of Fe is because FeCl_3_ was used as a catalyst in the synthesis of NGO@MnO_2_/PPy, which left traces of Fe. The elemental composition by the weight of Mn, O, N, C, and Fe are 65.1%, 18.1%, 1.7%, 6.7%, and 8.4%, respectively. From the above observations, we can say that the morphological structure provides convincing evidence for the introduction of PPy, MnO_2_, and NGO in the hybrid composite.

To study the synthesized composites further and to verify the presence of nanosized grains, FE-TEM analysis was conducted. Appendix A shows the structure of the NGO@MnO_2_ composite at different magnifications. It is evident from these figures that NGO@MnO_2_ consists of nanosized grains in the form of sheets and rods (see Appendix A). The stacked sheets of NGO@MnO_2_ have a thickness and diameter ranging from 10 to 20 nm and 30 to 40 nm, respectively.

Appendix A shows the structure of NGO@MnO_2_/PPy at different magnifications. It can be seen from these figures that the surface of NGO@MnO_2_/PPy is modified into spherical or disc-like structures with the introduction of PPy (see Appendix A). These figures also depict the presence of a hollow structure (see Appendix A) in the composite, which is favorable for the reversible gas adsorption/desorption [60]. Hence, with all these findings, the FE-TEM analysis proves the PPy insertion onto the NGO@MnO_2_ while forming the NGO@MnO_2_/PPy composite.

### 3.4. Experimental Result with a QCM Sensor

#### 3.4.1. Frequency Shift towards Different Concentrations of DMMP Vapor

The real time response of the QCM sensor while detecting DMMP is illustrated in Figure 5a,b. During the detection process, the instantaneous change in the frequency shift was perceived when the DMMP flowed inside the QCM chamber. It can be inferred from Figure 5 that NGO@MnO_2_ is sensitive to DMMP. We also notice that the addition of conducting polymer PPy increased the frequency shift of NGO@MnO_2_ dramatically due to the formation of a hydrogen bond between the target DMMP and PPy.

The hydrogen bond changes the charge distribution along the PPy backbone, affecting the transport of charge carriers. The resultant frequency shift is about six-times better at a DMMP concentration of 150 ppm. The literature also reports that the hydrogen atom containing the carboxyl group of PPy can couple non-covalently with the oxygen atom containing a phosphoryl group of DMMP (Figure 5c) through hydrogen bonding [61]. A process similar to this (Figure 5d) occurs when PPy reacts with the sarin. We also observed a fast recovery of the sensing material when nitrogen flowed inside the QCM chamber.

#### 3.4.2. Relationship between the Frequency Shift and Thickness of the Sensing Film

In order to analyze the effect of the thickness of the sensing material to the frequency shift of the QCM sensor, we first define the mass shift. The mass shift, ∆*m*, is defined as the mass of DMMP molecules adsorbed onto the surface of the sensing film during the adsorption process [62]. Appendix A shows the effect of mass accumulation of DMMP adsorbed onto the surface of the hybrid composites.

At a concentration of 25 ppm, ∆*m* in NGO@MnO_2_ and NGO@MnO_2_/PPy were 0.140 and 0.463 µg/cm^2^, respectively. At 125 ppm, the ∆m values were 0.367 and 1.315 µg/cm^2^, respectively. From this, we understand that the PPy-incorporated NGO@MnO_2_ adsorbed 3.58-times more DMMP than NGO@MnO_2_ at a concentration of 125 ppm. From Figure 5 and Appendix A, it appears that the frequency shift and mass accumulation are correlated [62].

#### 3.4.3. Selectivity and Polar Plot

In order to perform further studies on synthesized composite materials, the hybrid composites were subjected to DMMP, water and VOCs, such as ethanol, toluene, n-hexane, and methanol. From the results of this study given in Figure 6a,b, it can be observed that the sensing materials are highly sensitive to DMMP vapor rather than any potential vapors. In order to support this claim, we exposed the composites to toluene vapor with a concentration of 3522 ppm, which was more than 35-fold higher than that of DMMP.

Despite the elevated concentration of this potential interference, the frequency response of NGO@MnO_2_ and NGO@MnO_2_/PPy were more than 24-fold and 186-fold higher in DMMP than in toluene [63]. Hence, from this analysis, we conclude that the composite materials prepared possess an excellent frequency response towards CWAs—in particular, the nerve agent sarin.

We also draw the corresponding polar plots for studies conducted for the selectivity of the composites prepared. Figure 6c shows the polar plot response ratios (R1/R2 with R1 and R2 representing the responses of the NGO@MnO_2_/PPy-coated and NGO@MnO_2_-coated QCMs, respectively) when the composites are exposed to other vapors [64]. It is inferred from the study that the addition of PPy in NGO@MnO_2_ caused the minimum and maximum response ratios in toluene and ethanol, which are 1.06-times and 32.67-times greater, respectively.

In the case of water vapor and methanol, the response ratios were greater than 1.58 times and 2.68 times, respectively. These response ratios were greater than 14.60 times and 7.11 times, respectively, for N-hexane and DMMP. The addition of PPy greatly facilitates the hydrogen bond formation between DMMP molecules and the sensing film such that more DMMP is absorbed onto the surface enhancing the response of the sensing material [61].

#### 3.4.4. Linear Relationship between the Frequency Shift and Concentration of DMMP Vapor

We now move on to studying the linear relationship between the frequency shift and concentration of DMMP. Figure 7 shows the linearity of the QCM sensor for DMMP concentrations of 25 to 150 ppm. NGO@MnO_2_ and NGO@MnO_2_/PPy present in the QCM sensor have R^2^ values of 0.992 and 0.975, respectively. This is primarily due to good cohesive contact between the sensing materials and the electrodes as well as due to the large surface area [28] for DMMP gas detection.

The results also suggest that the composite materials developed can be promising candidates for practical applications [63]. Table 1 depicts the observed data of the calibration curve for the QCM sensor.

#### 3.4.5. Repeatability

To investigate the repeatability of the measurements, hybrid-composite-coated QCM sensors were exposed to a constant concentration of 25 ppm for four cycles of exposure and recovery. Appendix A shows the response of the QCM sensor for four successive cycles. The response curves show similar behaviors under the same conditions, although we noticed small drifts. These small drifts were presumed to be caused by differences in the concentration since the concentration of vapor cannot be controlled by the dilution process highly accurately [65].

The repeatability of each sensing materials in the QCM sensor was evaluated by calculating the coefficient of variation (*D*) of the repeated responses (n = 4) of the given hybrid composites when exposed to 25 ppm DMMP [66]. The *D* for the responses of the NGO@MnO_2_ and NGO@MnO_2_/PPy in the detection of 25 ppm DMMP was evaluated with the following equation:*D* (*%*) = *δ*/*k* × 100(5)
where *δ* is the standard deviation of a set of responses, and *k* is the average response of the QCM sensor coated with the hybrid composites when exposed to a DMMP vapor concentration of 25 ppm. The calculated values of *δ*, *k*, and *D* for NGO@MnO_2_ and NGO@MnO_2_/PPy are summarized in Table 2. In this table, a lower value of *D* indicates a higher repeatability [66]. We observe that NGO@MnO_2_ and NGO@MnO_2_/PPy show *D* values of 7.890% and 5.698%, respectively.

#### 3.4.6. Response and Recovery Times

The response times (*T*_90_) can be explained as the time taken to attain 90% of the equilibrium, whereas the recovery time is the time taken to reach 10% of the equilibrium via purging with the dry nitrogen [67]. Appendix A illustrates the response and recovery time of the hybrid composites. NGO@MnO_2_ exhibited response and recovery times of 96 and 116 s, respectively, in the detection of 75 ppm DMMP. On the other hand, NGO@MnO_2_/PPy showed response and recovery times of 103 and 117 s, respectively, in the detection of the same 75 ppm DMMP.

For both the composites, the recovery times were longer than the response times. The reversible response also indicates that the adsorption of DMMP onto the sensing materials consists of weak hydrogen bonding or Van der Waals forces [67,68]. The performance comparison was conducted with a composite-material-coated QCM sensor for the detection of DMMP as shown in Appendix A. This revealed that sensing materials provided excellent frequency response and shorter response times and recovery times in comparison with the other composite materials.

### 3.5. Experimental Results with a SAW Sensor

The performance of the SAW sensor was investigated at room temperature. Hence, we need to study the effect of oxygen in the sensing performance of the SAW sensor. We also note that atmospheric air contains 78.09% nitrogen, 20.95% oxygen, and 0.96% other gases, and hence studying the effect of oxygen is important. In our experiments, we used synthetic air to analyze the performance of the SAW sensor at a room temperature of 20 °C and 25–30% R.H.

#### 3.5.1. Frequency Shifts towards Different Concentrations of DMMP Vapor

We now study the frequency responses of the SAW sensor in this section. The real-time response of SAW sensors coated with the hybrid composites is shown in Figure 8. The SAW sensors studied were exposed to a DMMP concentration ranging from 25 to 150 ppm. In our experiments, one test cycle comprises a DMMP exposure of 5 min followed by a purging for 5 min using synthetic air.

At a concentration 25 ppm, the measured frequency shifts of NGO@MnO_2_ and NGO@MnO_2_/PPy were 145 and 98 Hz, respectively. Similarly, frequency shifts of 1114 and 1490 Hz were measured at 150 ppm of DMMP for the NGO@MnO_2_ and NGO@MnO_2_/PPy composites, respectively. The excellent measured response is attributed to the excellent sensitivities of manganese oxide, graphene oxide, and PPy to detect DMMP [37,69,70]. We also noticed a slight drift on the frequency shift that is due to the low desorption rate of DMMP molecules from the sensing layer surface [71].

#### 3.5.2. Selectivity and Polar Plot

To investigate the selectivity of SAW sensors coated with hybrid composites, they were exposed to DMMP, water, and VOCs at a fixed flow rate of 100 sccm. Figure 9a,b shows the logarithmic frequency shifts of SAW sensors when they are exposed to different vapors. From this response, it can be clearly seen that the hybrid composite materials synthesized possess higher frequency shift towards DMMP than the one it has for the interferences.

In order to understand this better, we look at a water vapor concentration that was 2892 ppm and 28-fold higher than that of the DMMP. Although the concentration of this potential interference was more, the frequency response of NGO@MnO_2_ and NGO@MnO_2_/PPy towards the DMMP was more than 91-fold and 141-fold greater, respectively, than the frequency response of water vapor [63]. Therefore, we conclude that the composite-material-deposited SAW sensor possessed excellent frequency responses towards the target CWA and simulant.

Figure 9c shows the polar plot response ratios (R3/R4) of the NGO@MnO_2_/PPy- and NGO@MnO_2_-coated SAW sensor when exposed to various vapors [64]. From this plot, we observe that in the SAW sensor incorporation of PPy into NGO@MnO_2_ increased the affinity to ethanol, which was 5.23-times greater than that of NGO@MnO_2_. In the case of DMMP, the R3/R4 of NGO@MnO_2_/PPy was 1.27-times greater than that of NGO@MnO_2_. The mechanism behind this increase is that, during the adsorption process, DMMP associates with the conductive polymer (PPy) to increase the number of free as well as mobile-hole charge carriers [70]. The R3/R4 values of various vapors is given in the following relation: R3/R4 of H_2_O > R3/R4 of Toluene > R3/R4 of *N*-hexane > R3/R4 of methanol. However, the response ratios of these vapors were less than 1.

#### 3.5.3. Linear Relationship between the Frequency Shift and Concentration of DMMP Vapor

Similar to the test conducted for the QCM sensor, we conducted a linearity test for the SAW sensor. Figure 10 illustrates the linearity of SAW sensors when exposed to a DMMP with a concentration ranging from 25 to 150 ppm. NGO@MnO_2_ and NGO@MnO_2_/PPy show *R*^2^ values of 0.979 and 0.989, respectively. From this figure, we can say that hybrid composite materials possess good linearity in the SAW sensors and reinforce the idea of being used for gas sensing [72]. Table 3 represents the observed data of the calibration curve for the SAW sensor.

#### 3.5.4. Repeatability

To investigate the repeatability of the sensing process of SAW sensors, the hybrid-composite-deposited SAW sensors were exposed to the same concentration of 75 ppm DMMP, and the corresponding responses are shown in Appendix A. In this study, one test cycle comprised 5 min of DMMP exposure followed by 5 min of purging with synthetic air.

Similar to the one in the QCM sensor, the repeatability of each sensing material in the SAW sensor was evaluated by calculating the *D* value of the repeated responses (n = 5) of the given hybrid composites when they are exposed to 75 ppm DMMP [66]. *D* for the responses of the hybrid composites was evaluated using Equation (5). The calculated values of *δ*, *k*, and *D* of the given hybrid-composite-coated SAW sensor are summarized in Table 4.

In this table, a lower value of *D* indicates higher repeatability [66]. From our calculations, we found that NGO@MnO_2_ and NGO@MnO_2_/PPy had *D* values of 11.3% and 11.6%, respectively. Since there were no significant changes in the response curve [73], we concluded that the composite materials developed possess excellent repeatability when they are used to detect DMMP. This repeatability also ensures that the fabricated composite materials can be used in practical applications that require a high level of repeatability in the sensing process [63].

#### 3.5.5. Response and Recovery Times

We now move on to studying the response and recovery times of the developed composite materials. The results are given in Appendix A, which illustrates the response and recovery times of the hybrid-composite-coated SAW sensor for an exposure of 75 ppm of DMMP. The response and recovery times of NGO@MnO_2_ were found to be 121 and 208 s, respectively, while those of NGO@MnO_2_/PPy were 120 and 197 s, respectively.

Similar to the QCM sensor, the recovery time was longer than the response time for the SAW sensors. This difference is attributed to the low desorption rate of DMMP molecules, which were initially adsorbed onto the sensing film surface [71]. We also conducted a performance comparison of the composite-material-coated SAW sensor for the detection of DMMP with other composite materials as shown in Appendix A. It can be inferred that the composite materials showed shorter response and recovery times compared with other composite materials.

### 3.6. Effect of R.H. on the Sensing Performance

To investigate the environmental effects on the performance of the sensor, we exposed the SAW sensor coated with NGO@MnO_2_/PPy to 25–30%, 60–65%, and 85–90% R.H. conditions at a constant temperature of 20 °C. The results of this analysis are shown in Appendix A. We previously, as part of our research, tested the QCM sensor for the detection of DMMP under 50% and 90% R.H. conditions at 20 °C but focus now on the SAW sensor in the present work.

From the results of this study, the frequency shifts of the NGO@MnO_2_/PPy-coated SAW sensor under 25–30%, 60–65%, and 85–90% R.H. conditions and 75 ppm were found to be 868, 828, and 543 Hz, respectively. The results for an exposure of 150 ppm were 1546, 1210, and 1190 Hz, respectively. For an exposure of 75 ppm, the dampings in frequency shift were about 4.60% and 34.42%, respectively, when the R.H. was changed from 25–30% to 60–65% and from 60–65% to 85–90%. Similarly, for an exposure of 150 ppm, and when the R.H. was changed from 25–30% to 60–65% and from 60–65% to 85–90%, an attenuation in the frequency shift occurred with values of 21.73% and 1.65%, respectively. This reduction in the frequency shift resulted from the adsorption of water molecules onto the thin sensing films, which failed to function properly as the R.H. increased to 100% [74].

### 3.7. Stability of Composite Materials

The stability test of NGO@MnO_2_/PPy-coated QCM sensor was conducted for the detection of 150 ppm DMMP for five consecutive days, which is illustrated in Appendix A. NGO@MnO_2_/PPy revealed satisfying stability results when comparing with the initial frequency shift for the detection of DMMP. Under normal room temperature, it was reported that graphene oxide exhibited only a small declination of 1.34% in the response for the detection of DMMP, which was monitored for 30 days [63].

In addition, manganese oxide-based composites were investigated for the long-term stability of 6 months. The composites revealed satisfying repeatability and stability with a slight declination in frequency shift of ~18 Hz [75]. Furthermore, it was reported that the polypyrrole-based composites showed more than 95% retention of the initial sensitivity during a monitoring period of 21 days [61].

### 3.8. Adsorption Mechanism of DMMP

Figure 11 shows the proposed adsorption mechanism of the DMMP onto the NGO@MnO_2_/PPy surface. DMMP may interact with the COOH group connected with the NGO, which indicates a strong interaction. Hydrogen bonds are formed due to the adsorption of DMMP between the P=O and hydroxyl groups (O–H–P=O) [76]. The surface selection rule [77] is also likely to occur in which hydroxyl groups are reoriented during the adsorption, leading to additional favorable perpendicular orientation of the OH bond with respect to the surface [76].

DMMP may be attached to the MnO_2_ surface by binding with a Bronsted acid site, and a Lewis acid site, which generates a bidendate structure to the nearest hydroxyl group and with the metal site [75]. In the case of the PPy, the phosphoryl group of DMMP binds with the carboxyl group, thereby, resulting in the formation of hydrogen bonds [61]. It is reported that the polymeric structure increases the hydrogen bonding strength in alcohols [78]. Thus, the NGO@MnO_2_/PPy composite showed increased adsorption to the DMMP.

## 4. Conclusions

As part of the present work, we developed and tested two hybrid composite materials that can act as a potential candidate for the detection of CWAs. Hydrothermal and thermal reduction processes were used to synthesize these composite materials. The hybrid composites were characterized using FT-IR, XRD, XPS, SEM, and TEM to study the elemental composition, intensities, energy differences, and morphological structures. The frequency shifts, selectivity, polar plots, linearity, repeatability, and response/recovery times for various concentrations of DMMP for both the QCM and SAW sensors were measured, plotted, and compared.

All these experiments were performed at a temperature of 20 °C and R.H. of 25–30%. The FTIR analysis confirmed the presence of elements used in the synthesis of the composite materials. The XRD results confirmed the tetragonal nanostructure of the composite materials. To add to this, a study using XPS exhibited the intensities and energy differences between two states of an element. The SEM and TEM results for the two composite materials also showed cauliflower-shaped morphology primarily due to the agglomeration of smaller particles.

In both the QCM and SAW sensors, NGO@MnO_2_/PPy showed enhanced frequency shifts when compared to the NGO@MnO_2_ composite. A selectivity test also showed that the composite materials were sensitive to DMMP rather than to any other potential interferences. The polar plots presented in this article show that the addition of PPy in NGO@MnO_2_ resulted in an enhanced frequency shift to DMMP and ethanol. Both the hybrid composite materials also exhibited excellent R^2^ when used in QCM and SAW sensors. Low values of *D* for both the composite materials confirmed their high repeatability in both sensors. We also noticed that the recovery times were longer than the response times in both the QCM and SAW sensors, which was mainly due to the low desorption rate.

## Figures and Tables

**Figure 1 nanomaterials-12-02965-f001:**
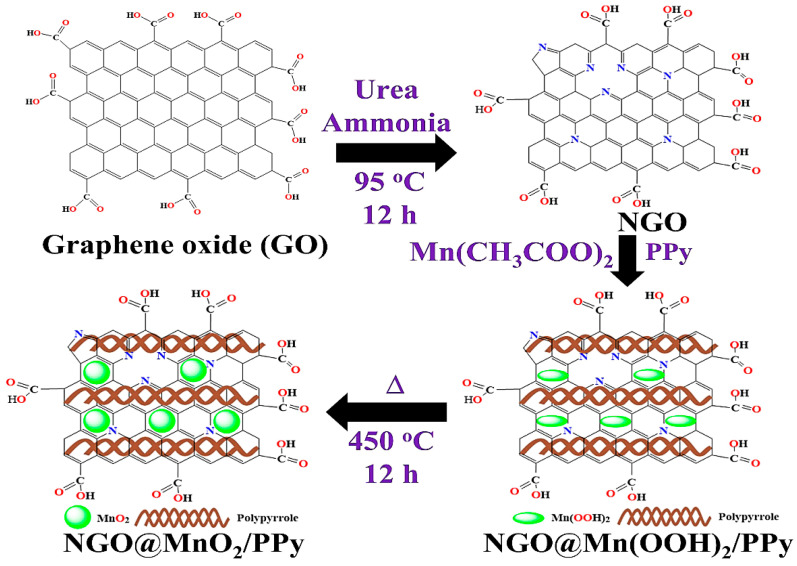
The synthesis process of NGO@MnO_2_/PPy.

**Figure 2 nanomaterials-12-02965-f002:**
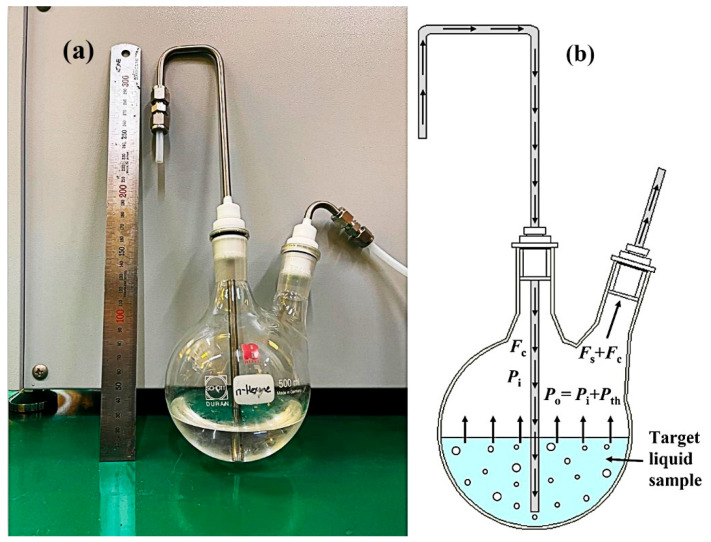
Vaporization system. (**a**) Bubbler flask used in the presented study. (**b**) Schematic diagram of the vapor generating process in the bubbler.

**Figure 3 nanomaterials-12-02965-f003:**
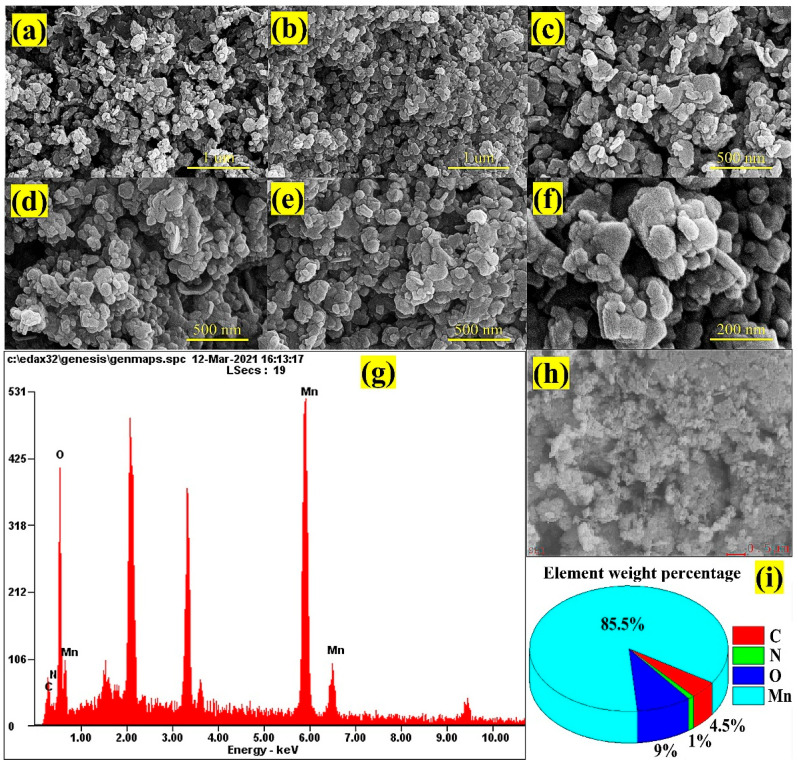
(**a**–**f**) High-resolution SEM images of NGO@MnO_2_ under different magnification levels. (**g**–**i**) EDX profile.

**Figure 4 nanomaterials-12-02965-f004:**
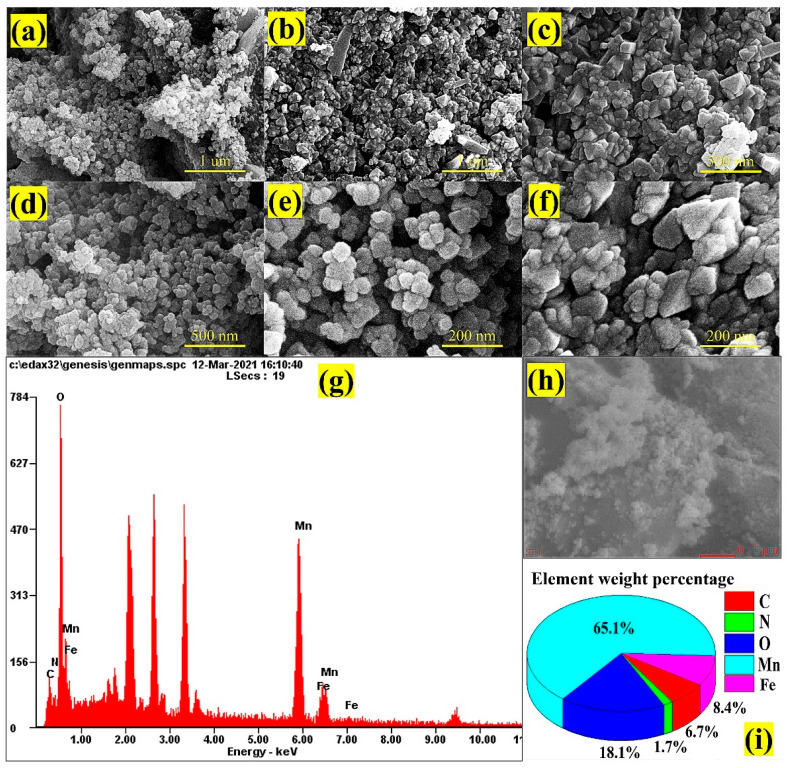
(**a**–**f**) High-resolution SEM images of NGO@MnO_2_/PPy under different magnification levels. (**g**–**i**) EDX profile.

**Figure 5 nanomaterials-12-02965-f005:**
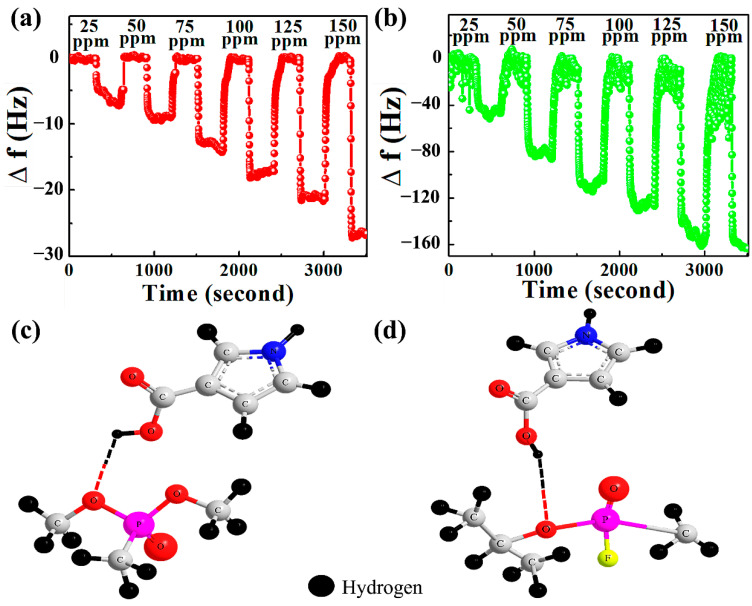
Frequency shifts of (**a**) NGO@MnO_2_, (**b**) NGO@MnO_2_/PPy in the QCM sensor (*T* = 20 °C, R.H. = 25–30%) and 3D computed graphics of hydrogen bond formation between (**c**) PPy and DMMP, and (**d**) PPy and Sarin. Figure (**c**,**d**) are not to scale.

**Figure 6 nanomaterials-12-02965-f006:**
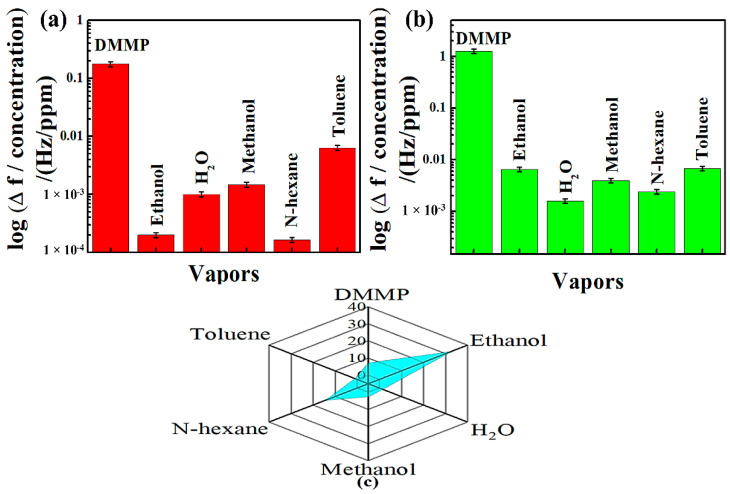
Selectivity of (**a**) NGO@MnO_2_, and (**b**) NGO@MnO_2_/PPy in the QCM sensor, and (**c**) the polar plot of response ratio (R1/R2) between the NGO@MnO_2_/PPy-coated and NGO@MnO_2_-coated QCM (*T* = 20 °C, R.H. = 25–30%).

**Figure 7 nanomaterials-12-02965-f007:**
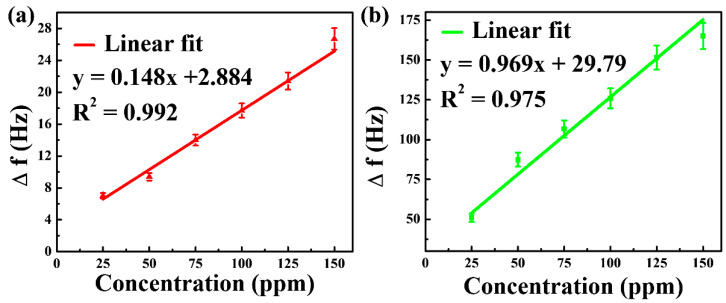
Linearity of (**a**) NGO@MnO_2_ and (**b**) NGO@MnO_2_/PPy in the QCM sensor (*T* = 20 °C, R.H. = 25–30%).

**Figure 8 nanomaterials-12-02965-f008:**
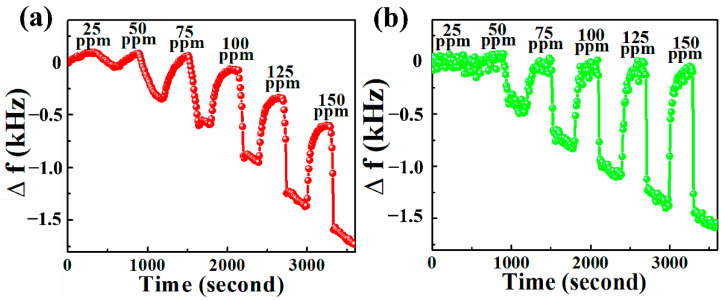
Frequency shifts of (**a**) NGO@MnO_2_ and (**b**) NGO@MnO_2_/PPy in the SAW sensor (*T* = 20 °C, R.H. = 25–30%).

**Figure 9 nanomaterials-12-02965-f009:**
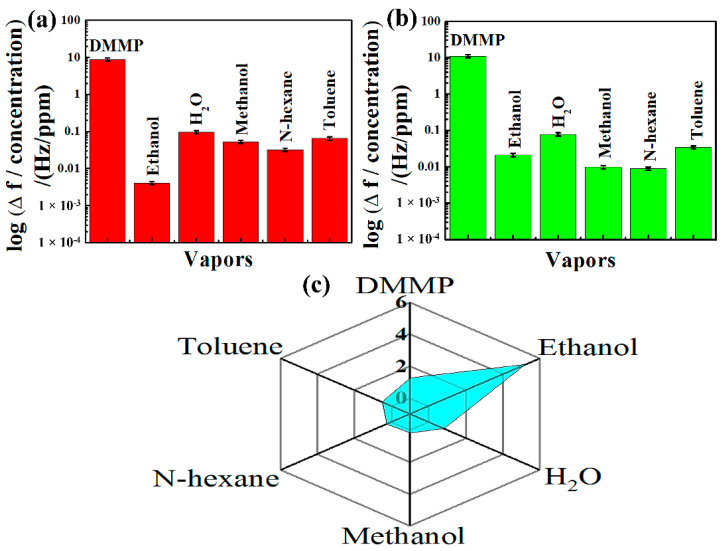
Selectivity of (**a**) NGO@MnO_2_, (**b**) NGO@MnO_2_/PPy in the SAW sensor, and (**c**) the polar plot of response ratios (R3/R4) between the NGO@MnO_2_/PPy-coated and NGO@MnO_2_-coated SAW (*T* = 20 °C, R.H. = 25–30%).

**Figure 10 nanomaterials-12-02965-f010:**
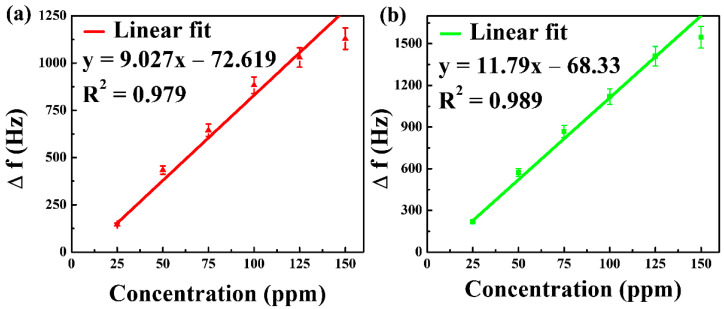
Linearity of (**a**) NGO@MnO_2_ and (**b**) NGO@MnO_2_/PPy in the SAW sensor (*T* = 20 °C, R.H. = 25–30%).

**Figure 11 nanomaterials-12-02965-f011:**
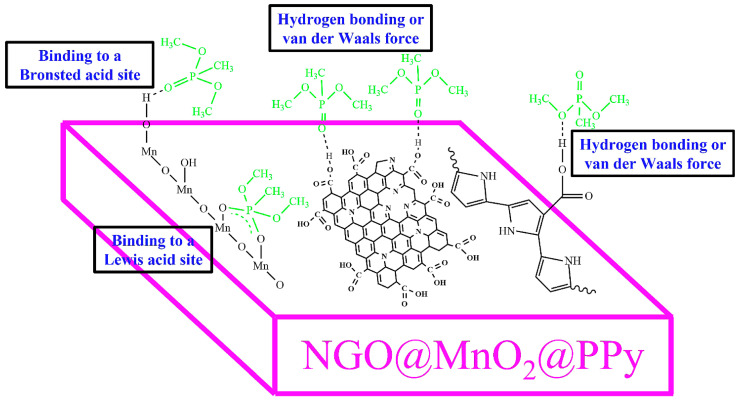
The proposed mechanism of DMMP adsorption on the NGO@MnO_2_/PPy composites with a Bronsted acid site, Lewis acid site, and hydrogen bonding or van der Waals force. The figure is not to scale.

**Table 1 nanomaterials-12-02965-t001:** Data of the calibration curve for the QCM sensor method.

	NGO@MnO_2_	NGO@MnO_2_/PPy
Regression equation	*y* = 0.148*x* + 2.884	*y* = 0.969*x* + 29.79
Standard error of the slope	0.009	0.068
Standard error of the intercept	0.554	4.492
Coefficient of determination (R^2^)	0.992	0.975
Number of data points	6	6

**Table 2 nanomaterials-12-02965-t002:** The calculated values of *δ*, *k*, and *D*.

	NGO@MnO_2_	NGO@MnO_2_/PPy
Standard deviation, *δ*	0.552 Hz	2.694 Hz
Average, *k*	7.000 Hz	47.293 Hz
Coefficient of variation, *D*	7.890%	5.698%

**Table 3 nanomaterials-12-02965-t003:** Data of the calibration curve for the SAW sensor method.

	NGO@MnO_2_	NGO@MnO_2_/PPy
Regression equation	*y* = 9.027*x* − 72.619	*y* = 11.79*x* − 68.33
Standard error of the slope	0.581	0.566
Standard error of the intercept	24.396	25.171
Coefficient of determination, R^2^	0.979	0.989
Number of data points	6	6

**Table 4 nanomaterials-12-02965-t004:** The calculated values of *δ*, *k*, and *D*.

	NGO@MnO_2_	NGO@MnO_2_/PPy
Standard deviation, *δ*	73.969 Hz	77.502 Hz
Average, *k*	656.370 Hz	665.834 Hz
Coefficient of variation, *D*	11.270%	11.640%

## Data Availability

Not applicable.

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
