# Peer review of "Nano-Sheet-like Morphology of Nitrogen-Doped Graphene-Oxide-Grafted Manganese Oxide and Polypyrrole Composite for Chemical Warfare Agent Simulant Detection"

_nanomaterials, 2022, doi:10.3390/nano12172965_

Round 1

Reviewer 1 Report

In this manuscript, the authors study the Nano Sheet Like Morphology of Nitrogen Doped Graphene Oxide Grafted Manganese Oxide and Polypyrrole Composite for Gas Sensor Applications.This work sounds very interesting and meaningful, and the analysis is reasonably clear, but there are still a few concerns regarding this article, therefore, I suggest it for publication in this journal after the following major revisions:

1. The long-term stability of the material needs to be measured.

2. The concentration of doping is also an important index affecting the sensing performance, and the influence of concentration should be studied.

3. Some grammatical errors are found. A proof-reading of the article will enhance the quality of the paper.

4. A list of acronyms is needed.

5. The diagram and illustration should be on one page.

Author Response

Reviewers' comments:1

Q1-1: The long-term stability of the material needs to be measured.

  • Response to reviewer’s: The authors would like to extend our thanks to the potential reviewer for valuable comments on this presented work. Long term stability is a crucial indicator used for the practical applications in gas monitoring. Under the normal room temperature, it has been reported that the graphene oxide exhibited only a small declination of 1.34% in the response for the detection of DMMP which was monitored for 30 days [related refrence: Wang, Y.; Yang, M.; Liu, W.; Dong, L.; Chen, D.; Peng, C. Gas Sensors Based on Assembled Porous Graphene Multilayer Frameworks for DMMP Detection. J. Mater. Chem. C 2019, 7, 9248–9256].

In addition, manganese oxide-based composites were investigated for the long-term stability of 6 months. The composites revealed satisfying repeatability and stability with slight declination in frequency shift of ~18 Hz [related reference: Thomas, G.; Spitzer, D. 3D Core–Shell TiO2@MnO2 Nanorod Arrays on Microcantilevers for Enhancing the Detection Sensitivity of Chemical Warfare Agents. ACS Appl. Mater. Interfaces 2021. 13(39) 47185-47197].

Furthermore, it has been reported that the polypyrrole based composites showed more than 95% retention of the initial sensitivity during the monitoring period of 21 days [related reference: Kwon, O.S.; Park, C.S.; Park, S.J.; Noh, S.; Kim, S.; Kong, H.J.; Bae, J.; Lee, C.-S.; Yoon, H. Carboxylic Acid-Functionalized Conducting-Polymer Nanotubes as Highly Sensitive Nerve-Agent Chemiresistors. Sci. Rep. 2016, 6, 1–7].

The potential reviewers suggested that the long-term stability needs to be measured. Given the time frame of 7 days to upload the revised manuscript, we conducted the stability test of the composite materials for 5 consecutive days. The results suggest that the composite materials showed satisfying stability for DMMP detection during the monitoring period. The results (diagram and explanation) can be found in the supplementary information and revised manuscript.

Figure. Stability test of the NGO@MnO2/PPy for detection of 150 ppm DMMP for five consecutive days.

Q1-2. The concentration of doping is also an important index affecting the sensing performance, and the influence of concentration should be studied.

  • Response to reviewer’s: Thank you very much for your comments on our manuscript. Generally, there are two ways to dope the graphene chemically: 1) organic molecules, metals, or gas adsorbed onto the surface of graphene; 2) substitutional doping which infuse the nitrogen atoms inside the lattice structure of graphene. When a nitrogen atom is introduced or doped into the graphene lattice structure, there occurs three different types of bonding configurations such as pyrrolic N, Pyridinic N, and graphitic N. The N doped graphene exhibits the distinct behavior than the graphene. The neighboring nitrogen dopants influences the spin density and carbon atoms charge distribution, that induces the activation region onto the surface of the graphene. After the nitrogen is doped onto the graphene monolayer, the Fermi level moves above the Dirac point, which results in the opening of the bandgap between the conduction and valence band. This bandgap makes nitrogen doped graphene to be used as a sensor.

It has been reported that increase in the nitrogen concentration into the graphene materials lead to the improvement in the sensing performance for dopamine, ascorbic acid, and uric acid [related reference: Megawati, M.; Chua, C.K.; Sofer, Z.; Klímová, K.; Pumera, M. Nitrogen-Doped Graphene: Effect of Graphite Oxide Precursors and Nitrogen Content on the Electrochemical Sensing Properties. Phys. Chem. Chem. Phys. 2017, 19, 15914–15923].

However, for the NADH, the sensitivity was decreased compared to the bare graphene surface. Also, the N-doped graphene oxide nanosheet greatly enhances the sensitivity towards the NO at the room temperature, indicating excellent potential as a gas sensor [related reference: Chang, Y.-S.; Chen, F.-K.; Tsai, D.-C.; Kuo, B.-H.; Shieu, F.-S. N-Doped Reduced Graphene Oxide for Room-Temperature NO Gas Sensors. Sci. Rep. 2021, 11, 1–12].

In case of the glucose biosensing, the concentration of nitrogen percentage achieved in the N-GO was from 0.11 to 1.35%, which showed excellent sensitivity and selectivity [Related refrence: Wang, Y.; Shao, Y.; Matson, D.W.; Li, J.; Lin, Y. Nitrogen-Doped Graphene and Its Application in Electrochemical Biosensing. ACS Nano 2010, 4, 1790–1798].

For sensing NO2, the composites constituting N-GO (6% concentration) showed optimum response and recovery times depicting suitable sensitivity and selectivity [Related reference: Badiezadeh, F.; Kimiagar, S. Modified WO3 Nanosheets by N-GO Nanocomposites to Form NO2 Sensor. J. Exp. Nanosci. 2021, 16, 144–158]. For the selective detection of CO, nitrogen doped graphene showed excellent response which would lead to the covalent bond between the lone pair electron and thus leading to vanishing of defective graphene system [Related reference: Ma, C.; Shao, X.; Cao, D. Nitrogen-Doped Graphene as an Excellent Candidate for Selective Gas Sensing. Sci. China Chem. 2014, 57, 911–917].

It was found that the doping temperature has huge impact on the concentration of nitrogen in the graphene oxide. The increase in doping temperature leads to the lower content of nitrogen and vice versa [Related reference: Song, J.; Kim, C.-M.; Yang, E.; Ham, M.-H.; Kim, I.S. The Effect of Doping Temperature on the Nitrogen-Bonding Configuration of Nitrogen-Doped Graphene by Hydrothermal Treatment. RSC Adv. 2017, 7, 20738–20741]. The electrical conductivity increases with the increase in the concentration of nitrogen in the graphene oxide [Related reference: Ngidi, N.P.D.; Ollengo, M.A.; Nyamori, V.O. Effect of Doping Temperatures and Nitrogen Precursors on the Physicochemical, Optical, and Electrical Conductivity Properties of Nitrogen-Doped Reduced Graphene Oxide. Materials (Basel). 2019, 12, 3376].   

Q1-3: Some grammatical errors are found. A proof-reading of the article will enhance the quality of the paper.

  • Response to reviewer’s: We would like to thank the potential reviewer for the comments on our manuscript. The potential reviewers suggested to check the grammatical errors in the manuscript which has been revised to our best efforts. We hope that the revised manuscript is enhanced to publish in the nanomaterials journal.

Q1-4: A list of acronyms is needed.

  • Response to reviewer’s: Thank you very much for your comments on our manuscript. The potential reviewers suggested to add a list of acronyms which has been added to the supplementary information (Table S1). The list of acronyms can also be found below:

Table. List of acronyms.

Acronyms

Definition

Acronyms

Definition

CWAs

Chemical warfare agents

IPA

Isopropyl alcohol

QCM

Quartz Crystal Microbalance

sccm

standard cubic centimeters per minute

SAW

Surface Acoustic Wave

MFCs

Mass Flow Controllers

DMMP

Dimethyl methyl phosphonate

ppm

parts per million

NGO

Nitrogen doped Graphene Oxide

Fc

Carrier gas flow rate

MnO2

Manganese dioxide

Po

Outlet pressure

PPy

Polypyrrole

Pth

Thermodynamic pressure

VOCs

Volatile Organic Compounds

Ps

Saturated vapor pressure

MEMS

Micro-electromechanical system

Fd

dilution gas flow rate

Co3O4

Cobalt oxide

SRS

Stanford Research System

R.H.

Relative Humidity

VNA

Vector Network Analyzer

ZnO

Zinc Oxide

SMA

SubMiniature Version A Connector

M.W.

Molecular weight

f

Resonant frequency

FTIR

Fourier Transform Infrared Spectroscopy

Cf

Sensitivity factor

XRD

X-Ray Diffraction

∆m

mass change

XPS

X-ray Photoelectron Spectroscopy

IDTs

Interdigital Transducers

SEM

Scanning Electron Microscopy

Fc

Central frequency

TEM

Transmission Electron Microscopy

v

Velocity

H2O2

Hydrogen peroxide

λ

Wavelength

DD

Double distilled water

FE-SEM

Field Emission Scanning Electron Microscopy

NaNO3

Sodium Nitrate

FE-TEM

Field Emission Transmission Electron Microscopy

H2SO4

Sulfuric acid

EDX

Energy Dispersive X-Ray Analysis

KOH

Potassium hydroxide

R2

Coefficient of determination

FeCl3

Ferric chloride

D

Coefficient of variation

KMnO4

Potassium permanganate

δ

Standard deviation (δ)

NH4OH

Ammonium hydroxide 

k

Average

PTFE

PTFE = Polytetrafluorethylene

T90

response time to attain 90 % of the equilibrium

Q1-5: The diagram and illustration should be on one page.

  • Response to reviewer’s: The authors would like to extend our thanks to the potential reviewer for valuable comments on this presented work. The potential reviewers suggested to include the diagram and the illustration on one page which has been updated in the revised manuscript as per the reviewer’s suggestions.

Reviewer 2 Report

The ms reports a work on nitrogen doped graphene oxide grafted MnO2 and Polypyrrole Composite for Gas Sensor Applications. Overall, the ms is interesting, however, a thorough revision is needed before reconsideration.

1) the title could be made a bit more catchy and gives the idea of the molecules being detected. For example the title should contain the word "chemical warfare agents".

2) The ms is far too long with an excessive number of references. I suggest to place some of the figures in the supporting info section;

3) Figs 11, 12, 16, 17, and 20 should report the error bars. How many times the experiments have been repeated?

4) Tabs 3 and 6 could go into the supporting info section;

5) 161 references are far too many. They should be reduced to less than 100;

6) The values of R2 in Figs 12 and 17 is quite low. is this acceptable? Wouldn't this mean that the as proposed sensors are not so good, or not enough reliable? 

Author Response

Reviewers' comments:2

Q2-1 : the title could be made a bit more catchy and gives the idea of the molecules being detected. For example the title should contain the word "chemical warfare agents".

  • Response to reviewer’s: The authors would like to thank the potential reviewer for the valuable comments on this presented work. The potential reviewers suggested to change the title to make it catchier. So, we have changed the title to “Nano sheet like morphology of nitrogen doped graphene oxide grafted manganese oxide and polypyrrole composite for chemical warfare agent’s simulant detection”. We hope that the revised title would best fit for our manuscript.

Q2-2. The ms is far too long with an excessive number of references. I suggest to place some of the figures in the supporting info section;

  • Response to reviewer’s: Thank you very much for your valuable comments on our manuscript. The potential reviewers suggested to place some figures into the supporting information. We have transferred the Figures like chemical structure of CWAs and simulant, FTIR and XRD pattern analysis of composite materials, FE-TEM images of composite materials, effect of mass accumulation of the composite materials, repeatability of composite materials in QCM sensor, response and recovery times of composite materials in QCM sensor, repeatability of composite materials in SAW sensor, response and recovery times of composite materials in SAW sensor, and effect of relative humidity on the sensing performance, to the supporting information section as per reviewer’s suggestions.

Q2-3. Figs 11, 12, 16, 17, and 20 should report the error bars. How many times the experiments have been repeated?

Response to reviewer’s: First of all, we would like to thank the potential reviewer for the comments on our manuscript. The potential reviewers suggested to report the error bars in Figure 11, 12, 16, 17, and 20. We have updated the error bars in the mentioned figure as per reviewer’s suggestion. The experiments have been repeated for four times.

Figure. Selectivity of (a) NGO@MnO2, and (b) NGO@MnO2/PPy in QCM sensor, (c) the polar plot of response ratio (R1/R2) between the NGO@MnO2/PPy coated and NGO@MnO2 coated QCM (T = 20 °C, R.H. = 25-30%).Figure. Linearity of (a) NGO@MnO2, and (b) NGO@MnO2/PPy in QCM sensor (T = 20 °C, R.H. = 25-30%).

Figure. Selectivity of (a) NGO@MnO2, and (b) NGO@MnO2/PPy in SAW sensor, (c) the polar plot of response ratios (R3/R4) between the NGO@MnO2/PPy coated and NGO@MnO2 coated SAW (T = 20 °C, R.H. = 25-30%).

Figure. Linearity of (a) NGO@MnO2, and (b) NGO@MnO2/PPy in SAW sensor (T = 20 °C, R.H. = 25-30%).

Figure. The effect of different R.H. conditions on the SAW sensor coated with NGO@MnO2/PPy under 25-30, 60-65, and 85-90% R.H. conditions at 20 °C for 75 and 150 ppm DMMP.

Q2-4. Tabs 3 and 6 could go into the supporting info section;

  • Response to reviewer’s: Thank you very much for your comments on our manuscript. The potential reviewers suggested to move Tables 3 and 6 to the supporting information section which has been transferred to the supporting information as per reviewer’s suggestion.

Q2-5. 161 references are far too many. They should be reduced to less than 100;

  • Response to reviewer’s: We would like to thank the potential reviewer for the valuable comment on our manuscript. The potential reviewers suggested to reduce the number of references to less than 100 which has been reduced to 98 in the revised manuscripts as per reviewer’s suggestion.

Q2-6. The values of R2 in Figs 12 and 17 is quite low. is this acceptable? Wouldn't this mean that the as proposed sensors are not so good, or not enough reliable?

  • Response to reviewer’s: The authors would like to thank the potential reviewer for the valuable comments on this presented work. Generally, the sensitivity of sensor for specific gases is complicated and challenging. Sensitivity is dependent on various parameters, such as gas adsorption and co-adsorption mechanisms, surface reaction kinetics, and electron transfer to or from the conduction band of the semiconductor. In this present study, which is focused on the DMMP sensor, linearity results seem to be low which may be due to the chemical properties of the composite materials, catalytic, and electronic effects on the dopants and surface modification of carbon nanomaterials. The conducting polymers such as polypyrrole (PPy), polyaniline, and polythiophene polymers have been widely used for the development of highly sensitive DMMP sensors. Many conducting polymers have demonstrated the changes in sensitivity on exposure to various gases and humidity. Amongst conducting polymers, PPy has been one of the most studied due to the easy deposition from aqueous and non-aqueous conditions using either chemical or electrochemical oxidation reactions. Another approach of PPy films have been used as a sensitive layer which is a p-type semiconductor whose main carriers are holes. Based on the properties, the PPy has been used as active layer for gas sensing and it has been proved that PPy is a promising material for gas sensor applications at room temperature conditions. Hence, the composite materials may have showed the different trend in linearity results. It is important to note that reported literatures [The related references are listed as below] whose R2 is relatively similar or lower than R2 obtained in our presented manuscript has been successfully applied to detect the DMMP and chemical warfare agents.

Related references

Zellers, E.T.; Pan, T.-S.; Patrash, S.J.; Han, M.; Batterman, S.A. Extended Disjoint Principal-Components Regression Analysis of SAW Vapor Sensor-Array Responses. Sensors Actuators B Chem. 1993, 12, 123–133.

Jonas, L.A.; Rehrmann, J.A. The Kinetics of Adsorption of Organo-Phosphorus Vapors from Air Mixtures by Activated Carbons. Carbon N. Y. 1972, 10, 657–663.

Sun, B.; Wang, Y.; Li, H.; Zhang, C.; Sun, S.; Sun, Y.; Ding, G. A Fast Response Breath Detection System Based on Bridge Tied CNT Sensor Fabricated by Micromachining. In Proceedings of the 2017 19th International Conference on Solid-State Sensors, Actuators and Microsystems (TRANSDUCERS); IEEE, 2017; pp. 1324–1327.

Sharma, P.K.; Gupta, G.; Singh, V. V; Tripathi, B.K.; Pandey, P.; Boopathi, M.; Singh, B.; Vijayaraghavan, R. Synthesis and Characterization of Polypyrrole by Cyclic Voltammetry at Different Scan Rate and Its Use in Electrochemical Reduction of the Simulant of Nerve Agents. Synth. Met. 2010, 160, 2631–2637.

O’Shea, K.E.; Aguila, A.; Vinodgopal, K.; Kamat, P. V Reaction Pathways and Kinetic Parameters of Sonolytically Induced Oxidation of Dimethyl Methylphosphonate in Air Saturated Aqueous Solutions. Res. Chem. Intermed. 1998, 24, 695–705.

Kim, J.; Park, H.; Kim, J.; Seo, B.-I.; Kim, J.-H. SAW Chemical Array Device Coated with Polymeric Sensing Materials for the Detection of Nerve Agents. Sensors 2020, 20, 7028.

Busmundrud, O. Vapour Breakthrough in Activated Carbon Beds. Carbon N. Y. 1993, 31, 279–286.

Munson, C.A.; De Lucia Jr, F.C.; Piehler, T.; McNesby, K.L.; Miziolek, A.W. Investigation of Statistics Strategies for Improving the Discriminating Power of Laser-Induced Breakdown Spectroscopy for Chemical and Biological Warfare Agent Simulants. Spectrochim. Acta Part B At. Spectrosc. 2005, 60, 1217–1224.

Huo, D.; Yang, L.; Hou, C.; Fa, H.; Luo, X.; Lu, Y.; Zheng, X.; Yang, J.; Yang, L. Molecular Interactions of Monosulfonate Tetraphenylporphyrin (TPPS1) and Meso-Tetra (4-Sulfonatophenyl) Porphyrin (TPPS) with Dimethyl Methylphosphonate (DMMP). Spectrochim. Acta Part A Mol. Biomol. Spectrosc. 2009, 74, 336–343.

Round 2

Reviewer 1 Report

The manuscript can be accepted in its present form.

Author Response

Response to Reviewer's and Academic Editor’s comment

Manuscript no.: nanomaterials-1851425

Title: Nano sheet like morphology of nitrogen doped graphene oxide grafted manganese oxide and polypyrrole composite for chemical warfare agent’s simulant detection

Dear Reviewers and Editor and Editorial office,

First of all, the authors appreciate the reviewers and academic editor for the most valuable comments. We have modified the manuscript according to the editor’s comment. Please find the detailed answers in the revised manuscript. We hope that the manuscript is now acceptable for publication in the nanomaterials (MDPI).

Reviewer’s  and Editor' comments:

  1. I agree with the Reviewer 2 that the reference list is too long. There are 98 refs even after revision. The authors should list the most related literature as reference. In addition, the information of some refs is incomplete, such as Ref. 94 and 98.

Response to Academic editor: The authors would like to extend our thanks to the editor for valuable comments on this presented work. We have only listed the most related literature as reference which has been reduced to 78 in the revised manuscript. The incomplete reference has been modified and completed in the revised manuscript. The modifications of references in 2nd revision are shown as below.

[Ref. No in the 1st rev] è [ New Ref. No in the 2nd rev.]

  1. -> [1] Kuča, K.; Pohanka, M. Chemical warfare agents. Mol. Clin. Environ. Toxicol. 2010, 543–558.
  2. ->[2] Tu, A.T. Basic information on nerve gas and the use of sarin by Aum Shinrikyo. J. Mass Spectrom. Soc. Jpn. 1996, 44, 293–320.
  3. ->[3] Sellström, A.; Cairns, S.; Barbeschi, M. United nations mission to investigate allegations of the use of chemical weapons in the Syrian Arab Republic. Final Rep. 2013, 12.

8 -> [4]. Gunzer, F.; Baether, W.; Zimmermann, S. Investigation of dimethyl methylphosphonate (DMMP) with an Ion mobility spectrometer using a pulsed electron source. Int. J. Ion Mobil. Spectrom. 2011, 14, 99–107.

4- > [5]. Brunol, E.; Berger, F.; Fromm, M.; Planade, R. Detection of dimethyl methylphosphonate (DMMP) by tin dioxide-based gas sensor: Response curve and understanding of the reactional mechanism. Sensors Actuators B Chem. 2006, 120, 35–41.

  1. Taranenko, N.; Alarie, J.; Stokes, D.L.; Vo‐Dinh, T. Surface‐Enhanced Raman Detection of Nerve Agent Simulant (DMMP and DIMP) Vapor on Electrochemically Prepared Silver Oxide Substrates. J. Raman Spectrosc. 1996, 27, 379–384.
  2. Tiwari, D.C.; Sharma, R.; Vyas, K.D.; Boopathi, M.; Singh, V. V; Pandey, P. Electrochemical incorporation of copper phthalocyanine in conducting polypyrrole for the sensing of DMMP. Sensors Actuators B Chem. 2010, 151, 256–264.
  3. Creasy, W.R.; Rodríguez, A.A.; Stuff, J.R.; Warren, R.W. Atomic emission detection for the quantitation of trimethylsilyl derivatives of chemical-warfare-agent related compounds in environmental samples. J. Chromatogr. A 1995, 709, 333–344.

9- >[ 6]. Lee, Y.-J.; Kim, J.-G.; Kim, J.-H.; Yun, J.; Jang, W.J. Detection of Dimethyl Methylphosphonate (DMMP) Using Polyhedral Oligomeric Silsesquioxane (POSS). J. Nanosci. Nanotechnol. 2018, 18, 6565–6569.

10->[7].    Kim, J.; Kim, E.; Kim, J.; Kim, J.; Ha, S. Four-Channel Monitoring System with Surface Acoustic Wave Sensors for Detection of Chemical Warfare Agents. 2020, 20, 7151–7157.

11->[8].    Tramonti, V.; Lofrumento, C.; Martina, M.R.; Lucchesi, G.; Caminati, G. Graphene Oxide/Silver Nanoparticles Platforms for the Detection and Discrimination of Native and Fibrillar Lysozyme: A Combined QCM and SERS Approach. Nanomaterials 2022, 12, 600.

12.->[9]    Hewa, T.M.P.; Tannock, G.A.; Mainwaring, D.E.; Harrison, S.; Fecondo, J. V The detection of influenza A and B viruses in clinical specimens using a quartz crystal microbalance. J. Virol. Methods 2009, 162, 14–21.

13->[10]. Koshets, I.A.; Kazantseva, Z.I.; Shirshov, Y.M.; Cherenok, S.A.; Kalchenko, V.I. Calixarene films as sensitive coatings for QCM-based gas sensors. Sensors Actuators B Chem. 2005, 106, 177–181.

14->[11]. Zampetti, E.; Macagnano, A.; Papa, P.; Bearzotti, A.; Petracchini, F.; Paciucci, L.; Pirrone, N. Exploitation of an integrated microheater on QCM sensor in particulate matter measurements. Sensors Actuators A Phys. 2017, 264, 205–211.

15->. [12] Buchatip, S.; Ananthanawat, C.; Sithigorngul, P.; Sangvanich, P.; Rengpipat, S.; Hoven, V.P. Detection of the shrimp pathogenic bacteria, Vibrio harveyi, by a quartz crystal microbalance-specific antibody based sensor. Sensors Actuators B Chem. 2010, 145, 259–264.

  1. ->[13] Zhu, Y.; Yuan, H.; Xu, J.; Xu, P.; Pan, Q. Highly stable and sensitive humidity sensors based on quartz crystal microbalance coated with hexagonal lamelliform monodisperse mesoporous silica SBA-15 thin film. Sensors Actuators B Chem. 2010, 144, 164–169.
  2. ->[14] Escuderos, M.E.; Sánchez, S.; Jiménez, A. Application of a quartz crystal microbalance (QCM) system coated with chromatographic adsorbents for the detection of olive oil volatile compounds. J. Sens. Technol. 2011, 1, 1.
  3. ->[15] Dirri, F.; Palomba, E.; Longobardo, A.; Zampetti, E.; Saggin, B.; Scaccabarozzi, D. A review of quartz crystal microbalances for space applications. Sensors Actuators A Phys. 2019, 287, 48–75.

19.-> [16] Huang, X.; Bai, Q.; Hu, J.; Hou, D. A practical model of quartz crystal microbalance in actual applications. Sensors 2017, 17, 1785.

20.-> [17] Miu, D.; Constantinoiu, I.; Enache, C.; Viespe, C. Effect of Pd/ZnO Morphology on Surface Acoustic Wave Sensor Response. Nanomaterials 2021, 11, 2598.

21.-> [18] Buff, W. SAW sensor system application. In Proceedings of the IEEE NTC, Conference Proceedings Microwave Systems Conference; IEEE, 1995; pp. 215–218.

  1. ->[19] Deng, J.; Zhang, R.; Wang, L.; Lou, Z.; Zhang, T. Enhanced sensing performance of the Co3O4 hierarchical nanorods to NH3 gas. Sensors Actuators B Chem. 2015, 209, 449–455.
  2. ->[20] Oberoi, A.; Sinha, R. A Novel MEMS based Surface Acoustic Wave Gas Sensor for Carbon Dioxide Detection in Hot-Process Areas. In Proceedings of the Proc. Int. Electron. Conf. Sensors Appl.; 2014; p. e001.
  3. ->[21] Wessa, T.; Rapp, M.; Sigrist, H. Immunosensing of photoimmobilized proteins on surface acoustic wave sensors. Colloids Surfaces B Biointerfaces 1999, 15, 139–146.
  4. ->[22] Inoue, Y.; Kato, Y.; Sato, K. Surface acoustic wave method for in situ determination of the amounts of enzyme–substrate complex formed on immobilized glucose oxidase during catalytic reaction. J. Chem. Soc. Faraday Trans. 1992, 88, 449–454.
  5. ->[23] Caliendo, C.; Verona, E.; Anisimkin, V.I. Surface acoustic wave humidity sensors: a comparison between different types of sensitive membrane. Smart Mater. Struct. 1997, 6, 707.
  6. ->[24] Heider, G. An Introduction to achieving Industrial Applications of Wireless Passive SAW Sensors for Advanced Monitoring. Proceeding-etc2014 2014, 14–17.
  7. ->[25] Li, Z.; Jones, Y.; Hossenlopp, J.; Cernosek, R.; Josse, F. Analysis of liquid-phase chemical detection using guided shear horizontal-surface acoustic wave sensors. Anal. Chem. 2005, 77, 4595–4603.
  8. Josse, F.; Bender, F.; Cernosek, R.W. Guided shear horizontal surface acoustic wave sensors for chemical and biochemical detection in liquids. Anal. Chem. 2001, 73, 5937–5944.
  9. ->[26] Vig, J.R.; Walls, F.L. A review of sensor sensitivity and stability. In Proceedings of the Proceedings of the 2000 IEEE/EIA International Frequency Control Symposium and Exhibition (Cat. No. 00CH37052); IEEE, 2000; pp. 30–33.
  10. ->[27] Bertoni, H.L.; Tamir, T. Unified theory of Rayleigh-angle phenomena for acoustic beams at liquid-solid interfaces. Appl. Phys. 1973, 2, 157–172.
  11. ->[28] Wang, Y.; Yang, Z.; Hou, Z.; Xu, D.; Wei, L.; Kong, E.S.W.; Zhang, Y. Flexible gas sensors with assembled carbon nanotube thin films for DMMP vapor detection. Sensors Actuators, B Chem. 2010, 150, 708–714.
  12. Wang, Y.; Zhou, Z.; Yang, Z.; Chen, X.; Xu, D.; Zhang, Y. Gas sensors based on deposited single-walled carbon nanotube networks for DMMP detection. Nanotechnology 2009, 20, 345502.
  13. ->[29] Gwizdz, P.; Radecka, M.; Zakrzewska, K. Array of chromium doped nanostructured TiO2 metal oxide gas sensors. Procedia Eng. 2014, 87, 1059–1062.
  14. Bai, J.; Zhou, B. Titanium dioxide nanomaterials for sensor applications. Chem. Rev. 2014, 114, 10131–10176.
  15. ->[30] Du, X.; Ying, Z.; Jiang, Y.; Liu, Z.; Yang, T.; Xie, G. Synthesis and evaluation of a new polysiloxane as SAW sensor coatings for DMMP detection. Sensors Actuators B Chem. 2008, 134, 409–413.
  16. ->[31] Li, H.-Y.; Zhao, S.-N.; Zang, S.-Q.; Li, J. Functional metal–organic frameworks as effective sensors of gases and volatile compounds. Chem. Soc. Rev. 2020, 49, 6364–6401.
  17. López‐Molino, J.; Amo‐Ochoa, P. Gas Sensors Based on Copper‐Containing Metal‐Organic Frameworks, Coordination Polymers, and Complexes. Chempluschem 2020, 85, 1564–1579.
  18. ->[32] Sharma, S.; Chauhan, P.; Husain, S. Liquefied petroleum gas sensor based on manganese (III) oxide and zinc manganese (III) oxide nanoparticles. Mater. Res. Express 2018, 5, 15014.
  19. Zhang, Q.; An, C.; Fan, S.; Shi, S.; Zhang, R.; Zhang, J.; Li, Q.; Zhang, D.; Hu, X.; Liu, J. Flexible gas sensor based on graphene/ethyl cellulose nanocomposite with ultra-low strain response for volatile organic compounds rapid detection. Nanotechnology 2018, 29, 285501.
  20. Zhai, Y.; Zhai, J.; Zhou, M.; Dong, S. Ordered magnetic core–manganese oxide shell nanostructures and their application in water treatment. J. Mater. Chem. 2009, 19, 7030–7035.
  21. Liu, X.; Chen, C.; Zhao, Y.; Jia, B. A review on the synthesis of manganese oxide nanomaterials and their applications on lithium-ion batteries. J. Nanomater. 2013, 2013.
  22. Chang, J.-K.; Lee, M.-T.; Tsai, W.-T. In situ Mn K-edge X-ray absorption spectroscopic studies of anodically deposited manganese oxide with relevance to supercapacitor applications. J. Power Sources 2007, 166, 590–594.
  23. Chen, H.; He, J.; Zhang, C.; He, H. Self-assembly of novel mesoporous manganese oxide nanostructures and their application in oxidative decomposition of formaldehyde. J. Phys. Chem. C 2007, 111, 18033–18038.
  24. ->[33] Zhao, Q.; Yan, Z.; Chen, C.; Chen, J. Spinels: controlled preparation, oxygen reduction/evolution reaction application, and beyond. Chem. Rev. 2017, 117, 10121–10211.
  25. Selampinar, F.; Toppare, L.; Akbulut, U.; Yalçin, T.; Süzer, Ş. A conducting composite of polypyrrole II. As a gas sensor. Synth. Met. 1995, 68, 109–116.

47.-> [34] An, K.H.; Jeong, S.Y.; Hwang, H.R.; Lee, Y.H. Enhanced sensitivity of a gas sensor incorporating single‐walled carbon nanotube–polypyrrole nanocomposites. Adv. Mater. 2004, 16, 1005–1009.

  1. ->[35] Pei, Z.; Ma, X.; Ding, P.; Zhang, W.; Luo, Z.; Li, G. Study of a QCM dimethyl methylphosphonate sensor based on a ZnO-modified nanowire-structured manganese dioxide film. Sensors 2010, 10, 8275–8290.
  2. ->[36] Ramesh, S.; Lee, Y.-J.; Msolli, S.; Kim, J.-G.; Kim, H.S.; Kim, J.-H. Synthesis of a Co 3 O 4@ gold/MWCNT/polypyrrole hybrid composite for DMMP detection in chemical sensors. RSC Adv. 2017, 7, 50912–50919.
  3. ->[37] Sayago, I.; Matatagui, D.; Fernández, M.J.; Fontecha, J.L.; Jurewicz, I.; Garriga, R.; Muñoz, E. Graphene oxide as sensitive layer in Love-wave surface acoustic wave sensors for the detection of chemical warfare agent simulants. Talanta 2016, 148, 393–400.

51.-> [38] Lavoie, J.; Srinivasan, S.; Nagarajan, R. Using cheminformatics to find simulants for chemical warfare agents. J. Hazard. Mater. 2011, 194, 85–91.

  1. ->[39] Sun, L.; Wang, L.; Tian, C.; Tan, T.; Xie, Y.; Shi, K.; Li, M.; Fu, H. Nitrogen-doped graphene with high nitrogen level via a one-step hydrothermal reaction of graphene oxide with urea for superior capacitive energy storage. Rsc Adv. 2012, 2, 4498–4506.
  2. Ramesh, S.; Yadav, H.M.; Shinde, S.K.; Bathula, C.; Lee, Y.-J.; Cheedarala, R.K.; Kim, H.-S.; Kim, H.S.; Kim, J.-H. Fabrication of nanostructured SnO2@ Co3O4/nitrogen doped graphene oxide composite for symmetric and asymmetric storage devices. J. Mater. Res. Technol. 2020, 9, 4183–4193.
  3. Lee, Y.-J.; Kim, G.-H.; Jung, D.; Kim, J.-H. Dimethyl Methylphosphonate Detection Using a Quartz Crystal Microbalance as Chemical Warfare Sensor. Nanosci. Nanotechnol. Lett. 2015, 7, 1015–1018.
  4. ->[40] Kim, E.; Kim, J.; Ha, S.; Song, C.; Kim, J.-H. Improved Performance of Surface Acoustic Wave Sensors by Plasma Treatments for Chemical Warfare Agents Monitoring. J. Nanosci. Nanotechnol. 2020, 20, 7145–7150.
  5. ->[41] Hersee, S.D.; Ballingall, J.M. The operation of metalorganic bubblers at reduced pressure. J. Vac. Sci. Technol. A Vacuum, Surfaces, Film. 1990, 8, 800–804.
  6. ->[42] Dean, J.A. Lange’s handbook of chemistry. Mater. Manuf. Process 1990, 5, 687–688.
  7. ->[43] Kim, Y.S.; Ha, S.C.; Yang, H.; Kim, Y.T. Gas sensor measurement system capable of sampling volatile organic compounds (VOCs) in wide concentration range. Sensors Actuators, B Chem. 2007, 122, 211–218.
  8. ->[44] Standford Research systems QCM200 Digital controller: Operation and Service Manual; Sunnyvale, CA, USA, 2011;
  9. ->[45] Curie, J.; Curie, P. An oscillating quartz crystal mass detector. Rendu 1880, 91, 294–297.
  10. ->[46] Sauerbrey, G. Verwendung von Schwingquarzen zur Wägung dünner Schichten und zur Mikrowägung. Zeitschrift für Phys. 1959, 155, 206–222.
  11. ->[47] Rayleigh, Lord On waves propagated along the plane surface of an elastic solid. Proc. London Math. Soc. 1885, 1, 4–11.

64.-> [48] White, R.M.; Voltmer, F.W. Direct piezoelectric coupling to surface elastic waves. Appl. Phys. Lett. 1965, 7, 314–316.

  1. ->[49] Benes, E.; Groschl, M.; Seifert, F.; Pohl, A. Comparison between BAW and SAW sensor principles. IEEE Trans. Ultrason. Ferroelectr. Freq. Control 1998, 45, 1314–1330.
  2. ->[50] Kim, J.; Park, H.; Kim, J.; Seo, B.-I.; Kim, J.-H. SAW Chemical Array Device Coated with Polymeric Sensing Materials for the Detection of Nerve Agents. Sensors 2020, 20, 7028.

66.-> [51] Ramesh, S.; Haldorai, Y.; Kim, H.S.; Kim, J.-H. A nanocrystalline Co 3 O 4@ polypyrrole/MWCNT hybrid nanocomposite for high performance electrochemical supercapacitors. RSC Adv. 2017, 7, 36833–36843.

  1. ->[52] Zheng, M.; Zhang, H.; Gong, X.; Xu, R.; Xiao, Y.; Dong, H.; Liu, X.; Liu, Y. A simple additive-free approach for the synthesis of uniform manganese monoxide nanorods with large specific surface area. Nanoscale Res. Lett. 2013, 8, 1–7.
  2. Chen, H.; He, J. Facile synthesis of monodisperse manganese oxide nanostructures and their application in water treatment. J. Phys. Chem. C 2008, 112, 17540–17545.
  3. ->[53] Šetka, M.; Drbohlavová, J.; Hubálek, J. Nanostructured polypyrrole-based ammonia and volatile organic compound sensors. Sensors 2017, 17, 562.
  4. ->[54] Ramesh, S.; Yadav, H.M.; Karuppasamy, K.; Vikraman, D.; Kim, H.-S.; Kim, J.-H.; Kim, H.S. Fabrication of manganese oxide@ nitrogen doped graphene oxide/polypyrrole (MnO2@ NGO/PPy) hybrid composite electrodes for energy storage devices. J. Mater. Res. Technol. 2019, 8, 4227–4238.
  5. ->[55] Huang, M.; Li, F.; Dong, F.; Zhang, Y.X.; Zhang, L.L. MnO 2-based nanostructures for high-performance supercapacitors. J. Mater. Chem. A 2015, 3, 21380–21423.
  6. ->[56] Bai, X.P.; Zhao, X.; Fan, W.L. Preparation and enhanced photocatalytic hydrogen-evolution activity of ZnGa 2 O 4/N-rGO heterostructures. RSC Adv. 2017, 7, 53145–53156.
  7. Ramesh, S.; Karuppasamy, K.; Kim, H.-S.; Kim, H.S.; Kim, J.-H. Hierarchical Flowerlike 3D nanostructure of Co 3 O 4@ MnO 2/N-doped Graphene oxide (NGO) hybrid composite for a high-performance supercapacitor. Sci. Rep. 2018, 8, 1–11.
  8. ->[57] Wahid, M.; Parte, G.; Phase, D.; Ogale, S. Yogurt: a novel precursor for heavily nitrogen doped supercapacitor carbon. J. Mater. Chem. A 2015, 3, 1208–1215.
  9. ->[58] Yoon, H.; Jang, J. Conducting‐polymer nanomaterials for high‐performance sensor applications: issues and challenges. Adv. Funct. Mater. 2009, 19, 1567–1576.
  10. Hatchett, D.W.; Josowicz, M. Composites of intrinsically conducting polymers as sensing nanomaterials. Chem. Rev. 2008, 108, 746–769.

77.-> [59] Geng, L.; Wu, S. Preparation, characterization and gas sensitivity of polypyrrole/γ-Fe2O3 hybrid materials. Mater. Res. Bull. 2013, 48, 4339–4343.

  1. Hamilton, S.; Hepher, M.J.; Sommerville, J. Polypyrrole materials for detection and discrimination of volatile organic compounds. Sensors Actuators B Chem. 2005, 107, 424–432.
  2. ->[60] Joulazadeh, M.; Navarchian, A.H. Ammonia detection of one-dimensional nano-structured polypyrrole/metal oxide nanocomposites sensors. Synth. Met. 2015, 210, 404–411.
  3. ->[61] Kwon, O.S.; Park, C.S.; Park, S.J.; Noh, S.; Kim, S.; Kong, H.J.; Bae, J.; Lee, C.-S.; Yoon, H. Carboxylic acid-functionalized conducting-polymer nanotubes as highly sensitive nerve-agent chemiresistors. Sci. Rep. 2016, 6, 1–7.
  4. ->[62] Gupta, V.K.; Yola, M.L.; Eren, T.; Atar, N. Selective QCM sensor based on atrazine imprinted polymer: its application to wastewater sample. Sensors Actuators B Chem. 2015, 218, 215–221.
  5. ->[63] Wang, Y.; Yang, M.; Liu, W.; Dong, L.; Chen, D.; Peng, C. Gas sensors based on assembled porous graphene multilayer frameworks for DMMP detection. J. Mater. Chem. C 2019, 7, 9248–9256.

83.-> [64] Chen, D.; Zhang, K.; Zhou, H.; Fan, G.; Wang, Y.; Li, G.; Hu, R. A wireless-electrodeless quartz crystal microbalance with dissipation DMMP sensor. Sensors Actuators B Chem. 2018, 261, 408–417.

  1. ->[65] He, W.; Liu, Z.; Du, X.; Jiang, Y.; Xiao, D. Analytical application of poly {methyl [3-(2-hydroxy-3, 4-difluoro) phenyl] propyl siloxane} as a QCM coating for DMMP detection. Talanta 2008, 76, 698–702.
  2. ->[66] Alizadeh, T.; Soltani, L.H. Reduced graphene oxide-based gas sensor array for pattern recognition of DMMP vapor. Sensors Actuators B Chem. 2016, 234, 361–370.
  3. ->[67] Lama, S.; Kim, J.; Ramesh, S.; Lee, Y.-J.; Kim, J.; Kim, J.-H. Highly Sensitive Hybrid Nanostructures for Dimethyl Methyl Phosphonate Detection. Micromachines 2021, 12, 648.
  4. ->[68] Haghighi, E.; Zeinali, S. Nanoporous MIL-101 (Cr) as a sensing layer coated on a quartz crystal microbalance (QCM) nanosensor to detect volatile organic compounds (VOCs). RSC Adv. 2019, 9, 24460–24470.
  5. ->[69] Segal, S.R.; Suib, S.L.; Tang, X.; Satyapal, S. Photoassisted decomposition of dimethyl methylphosphonate over amorphous manganese oxide catalysts. Chem. Mater. 1999, 11, 1687–1695.
  6. ->[70] Collins, G.E.; Buckley, L.J. Conductive polymer-coated fabrics for chemical sensing. Synth. Met. 1996, 78, 93–101.
  7. ->[71] Chevallier, E.; Scorsone, E.; Bergonzo, P. New sensitive coating based on modified diamond nanoparticles for chemical SAW sensors. Sensors Actuators B Chem. 2011, 154, 238–244.
  8. ->[72] Hu, N.; Wang, Y.; Chai, J.; Gao, R.; Yang, Z.; Kong, E.S.-W.; Zhang, Y. Gas sensor based on p-phenylenediamine reduced graphene oxide. Sensors Actuators B Chem. 2012, 163, 107–114.
  9. ->[73] Wang, Y.; Du, X.; Li, Y.; Long, Y.; Qiu, D.; Tai, H.; Tang, X.; Jiang, Y. A simple route to functionalize siloxane polymers for DMMP sensing. J. Appl. Polym. Sci. 2013, 130, 4516–4520.
  10. ->[74] Du, X.; Wang, Z.; Huang, J.; Tao, S.; Tang, X.; Jiang, Y. A new polysiloxane coating on QCM sensor for DMMP vapor detection. J. Mater. Sci. 2009, 44, 5872–5876.
  11. ->[75] Thomas, G.; Spitzer, D. 3D Core–Shell TiO2@ MnO2 Nanorod Arrays on Microcantilevers for Enhancing the Detection Sensitivity of Chemical Warfare Agents. ACS Appl. Mater. Interfaces 2021.
  12. ->[76] Bertilsson, L.; Potje-Kamloth, K.; Liess, H.-D.; Engquist, I.; Liedberg, B. Adsorption of dimethyl methylphosphonate on self-assembled alkanethiolate monolayers. J. Phys. Chem. B 1998, 102, 1260–1269.
  13. ->[77] Greenler, R.G. Infrared study of adsorbed molecules on metal surfaces by reflection techniques. J. Chem. Phys. 1966, 44, 310–315.
  14. Francis, S.; Ellison, A.H. Infrared spectra of monolayers on metal mirrors. JOSA 1959, 49, 131–138.
  15. ->[78] Bellamy, L.J. The IR-Spectra of Complex Molecules, vol. 205 1975.

Reviewer 2 Report

The authors have revised the ms taking into account the referees' suggestions. I think the ms is now suitable for publication.

Author Response

Response to the Reviewer’s and Academic Editor’s comment

Manuscript no.: nanomaterials-1851425

Title: Nano sheet like morphology of nitrogen doped graphene oxide grafted manganese oxide and polypyrrole composite for chemical warfare agent’s simulant detection

Dear Reviewers and Editor and Editorial office,

First of all, the authors appreciate the reviewers and the academic editor for the most valuable comments. We have modified the manuscript according to the editor’s comment. Please find the detailed answers in the revised manuscript. We hope that the manuscript is now acceptable for publication in the nanomaterials (MDPI).

Reviewer’s  and Editor' comments:

  1. I agree with the Reviewer 2 that the reference list is too long. There are 98 refs even after revision. The authors should list the most related literature as reference. In addition, the information of some refs is incomplete, such as Ref. 94 and 98.

Response to Academic editor: The authors would like to extend our thanks to the editor for valuable comments on this presented work. We have only listed the most related literature as reference which has been reduced to 78 in the revised manuscript. The incomplete reference has been modified and completed in the revised manuscript. The modifications of references in 2nd revision are shown as below.

[Ref. No in the 1st rev] è [ New Ref. No in the 2nd rev.]

  1. -> [1] Kuča, K.; Pohanka, M. Chemical warfare agents. Mol. Clin. Environ. Toxicol. 2010, 543–558.
  2. ->[2] Tu, A.T. Basic information on nerve gas and the use of sarin by Aum Shinrikyo. J. Mass Spectrom. Soc. Jpn. 1996, 44, 293–320.
  3. ->[3] Sellström, A.; Cairns, S.; Barbeschi, M. United nations mission to investigate allegations of the use of chemical weapons in the Syrian Arab Republic. Final Rep. 2013, 12.

8 -> [4]. Gunzer, F.; Baether, W.; Zimmermann, S. Investigation of dimethyl methylphosphonate (DMMP) with an Ion mobility spectrometer using a pulsed electron source. Int. J. Ion Mobil. Spectrom. 2011, 14, 99–107.

4- > [5]. Brunol, E.; Berger, F.; Fromm, M.; Planade, R. Detection of dimethyl methylphosphonate (DMMP) by tin dioxide-based gas sensor: Response curve and understanding of the reactional mechanism. Sensors Actuators B Chem. 2006, 120, 35–41.

  1. Taranenko, N.; Alarie, J.; Stokes, D.L.; Vo‐Dinh, T. Surface‐Enhanced Raman Detection of Nerve Agent Simulant (DMMP and DIMP) Vapor on Electrochemically Prepared Silver Oxide Substrates. J. Raman Spectrosc. 1996, 27, 379–384.
  2. Tiwari, D.C.; Sharma, R.; Vyas, K.D.; Boopathi, M.; Singh, V. V; Pandey, P. Electrochemical incorporation of copper phthalocyanine in conducting polypyrrole for the sensing of DMMP. Sensors Actuators B Chem. 2010, 151, 256–264.
  3. Creasy, W.R.; Rodríguez, A.A.; Stuff, J.R.; Warren, R.W. Atomic emission detection for the quantitation of trimethylsilyl derivatives of chemical-warfare-agent related compounds in environmental samples. J. Chromatogr. A 1995, 709, 333–344.

9- >[ 6]. Lee, Y.-J.; Kim, J.-G.; Kim, J.-H.; Yun, J.; Jang, W.J. Detection of Dimethyl Methylphosphonate (DMMP) Using Polyhedral Oligomeric Silsesquioxane (POSS). J. Nanosci. Nanotechnol. 2018, 18, 6565–6569.

10->[7].    Kim, J.; Kim, E.; Kim, J.; Kim, J.; Ha, S. Four-Channel Monitoring System with Surface Acoustic Wave Sensors for Detection of Chemical Warfare Agents. 2020, 20, 7151–7157.

11->[8].    Tramonti, V.; Lofrumento, C.; Martina, M.R.; Lucchesi, G.; Caminati, G. Graphene Oxide/Silver Nanoparticles Platforms for the Detection and Discrimination of Native and Fibrillar Lysozyme: A Combined QCM and SERS Approach. Nanomaterials 2022, 12, 600.

12.->[9]    Hewa, T.M.P.; Tannock, G.A.; Mainwaring, D.E.; Harrison, S.; Fecondo, J. V The detection of influenza A and B viruses in clinical specimens using a quartz crystal microbalance. J. Virol. Methods 2009, 162, 14–21.

13->[10]. Koshets, I.A.; Kazantseva, Z.I.; Shirshov, Y.M.; Cherenok, S.A.; Kalchenko, V.I. Calixarene films as sensitive coatings for QCM-based gas sensors. Sensors Actuators B Chem. 2005, 106, 177–181.

14->[11]. Zampetti, E.; Macagnano, A.; Papa, P.; Bearzotti, A.; Petracchini, F.; Paciucci, L.; Pirrone, N. Exploitation of an integrated microheater on QCM sensor in particulate matter measurements. Sensors Actuators A Phys. 2017, 264, 205–211.

15->. [12] Buchatip, S.; Ananthanawat, C.; Sithigorngul, P.; Sangvanich, P.; Rengpipat, S.; Hoven, V.P. Detection of the shrimp pathogenic bacteria, Vibrio harveyi, by a quartz crystal microbalance-specific antibody based sensor. Sensors Actuators B Chem. 2010, 145, 259–264.

  1. ->[13] Zhu, Y.; Yuan, H.; Xu, J.; Xu, P.; Pan, Q. Highly stable and sensitive humidity sensors based on quartz crystal microbalance coated with hexagonal lamelliform monodisperse mesoporous silica SBA-15 thin film. Sensors Actuators B Chem. 2010, 144, 164–169.
  2. ->[14] Escuderos, M.E.; Sánchez, S.; Jiménez, A. Application of a quartz crystal microbalance (QCM) system coated with chromatographic adsorbents for the detection of olive oil volatile compounds. J. Sens. Technol. 2011, 1, 1.
  3. ->[15] Dirri, F.; Palomba, E.; Longobardo, A.; Zampetti, E.; Saggin, B.; Scaccabarozzi, D. A review of quartz crystal microbalances for space applications. Sensors Actuators A Phys. 2019, 287, 48–75.

19.-> [16] Huang, X.; Bai, Q.; Hu, J.; Hou, D. A practical model of quartz crystal microbalance in actual applications. Sensors 2017, 17, 1785.

20.-> [17] Miu, D.; Constantinoiu, I.; Enache, C.; Viespe, C. Effect of Pd/ZnO Morphology on Surface Acoustic Wave Sensor Response. Nanomaterials 2021, 11, 2598.

21.-> [18] Buff, W. SAW sensor system application. In Proceedings of the IEEE NTC, Conference Proceedings Microwave Systems Conference; IEEE, 1995; pp. 215–218.

  1. ->[19] Deng, J.; Zhang, R.; Wang, L.; Lou, Z.; Zhang, T. Enhanced sensing performance of the Co3O4 hierarchical nanorods to NH3 gas. Sensors Actuators B Chem. 2015, 209, 449–455.
  2. ->[20] Oberoi, A.; Sinha, R. A Novel MEMS based Surface Acoustic Wave Gas Sensor for Carbon Dioxide Detection in Hot-Process Areas. In Proceedings of the Proc. Int. Electron. Conf. Sensors Appl.; 2014; p. e001.
  3. ->[21] Wessa, T.; Rapp, M.; Sigrist, H. Immunosensing of photoimmobilized proteins on surface acoustic wave sensors. Colloids Surfaces B Biointerfaces 1999, 15, 139–146.
  4. ->[22] Inoue, Y.; Kato, Y.; Sato, K. Surface acoustic wave method for in situ determination of the amounts of enzyme–substrate complex formed on immobilized glucose oxidase during catalytic reaction. J. Chem. Soc. Faraday Trans. 1992, 88, 449–454.
  5. ->[23] Caliendo, C.; Verona, E.; Anisimkin, V.I. Surface acoustic wave humidity sensors: a comparison between different types of sensitive membrane. Smart Mater. Struct. 1997, 6, 707.
  6. ->[24] Heider, G. An Introduction to achieving Industrial Applications of Wireless Passive SAW Sensors for Advanced Monitoring. Proceeding-etc2014 2014, 14–17.
  7. ->[25] Li, Z.; Jones, Y.; Hossenlopp, J.; Cernosek, R.; Josse, F. Analysis of liquid-phase chemical detection using guided shear horizontal-surface acoustic wave sensors. Anal. Chem. 2005, 77, 4595–4603.
  8. Josse, F.; Bender, F.; Cernosek, R.W. Guided shear horizontal surface acoustic wave sensors for chemical and biochemical detection in liquids. Anal. Chem. 2001, 73, 5937–5944.
  9. ->[26] Vig, J.R.; Walls, F.L. A review of sensor sensitivity and stability. In Proceedings of the Proceedings of the 2000 IEEE/EIA International Frequency Control Symposium and Exhibition (Cat. No. 00CH37052); IEEE, 2000; pp. 30–33.
  10. ->[27] Bertoni, H.L.; Tamir, T. Unified theory of Rayleigh-angle phenomena for acoustic beams at liquid-solid interfaces. Appl. Phys. 1973, 2, 157–172.
  11. ->[28] Wang, Y.; Yang, Z.; Hou, Z.; Xu, D.; Wei, L.; Kong, E.S.W.; Zhang, Y. Flexible gas sensors with assembled carbon nanotube thin films for DMMP vapor detection. Sensors Actuators, B Chem. 2010, 150, 708–714.
  12. Wang, Y.; Zhou, Z.; Yang, Z.; Chen, X.; Xu, D.; Zhang, Y. Gas sensors based on deposited single-walled carbon nanotube networks for DMMP detection. Nanotechnology 2009, 20, 345502.
  13. ->[29] Gwizdz, P.; Radecka, M.; Zakrzewska, K. Array of chromium doped nanostructured TiO2 metal oxide gas sensors. Procedia Eng. 2014, 87, 1059–1062.
  14. Bai, J.; Zhou, B. Titanium dioxide nanomaterials for sensor applications. Chem. Rev. 2014, 114, 10131–10176.
  15. ->[30] Du, X.; Ying, Z.; Jiang, Y.; Liu, Z.; Yang, T.; Xie, G. Synthesis and evaluation of a new polysiloxane as SAW sensor coatings for DMMP detection. Sensors Actuators B Chem. 2008, 134, 409–413.
  16. ->[31] Li, H.-Y.; Zhao, S.-N.; Zang, S.-Q.; Li, J. Functional metal–organic frameworks as effective sensors of gases and volatile compounds. Chem. Soc. Rev. 2020, 49, 6364–6401.
  17. López‐Molino, J.; Amo‐Ochoa, P. Gas Sensors Based on Copper‐Containing Metal‐Organic Frameworks, Coordination Polymers, and Complexes. Chempluschem 2020, 85, 1564–1579.
  18. ->[32] Sharma, S.; Chauhan, P.; Husain, S. Liquefied petroleum gas sensor based on manganese (III) oxide and zinc manganese (III) oxide nanoparticles. Mater. Res. Express 2018, 5, 15014.
  19. Zhang, Q.; An, C.; Fan, S.; Shi, S.; Zhang, R.; Zhang, J.; Li, Q.; Zhang, D.; Hu, X.; Liu, J. Flexible gas sensor based on graphene/ethyl cellulose nanocomposite with ultra-low strain response for volatile organic compounds rapid detection. Nanotechnology 2018, 29, 285501.
  20. Zhai, Y.; Zhai, J.; Zhou, M.; Dong, S. Ordered magnetic core–manganese oxide shell nanostructures and their application in water treatment. J. Mater. Chem. 2009, 19, 7030–7035.
  21. Liu, X.; Chen, C.; Zhao, Y.; Jia, B. A review on the synthesis of manganese oxide nanomaterials and their applications on lithium-ion batteries. J. Nanomater. 2013, 2013.
  22. Chang, J.-K.; Lee, M.-T.; Tsai, W.-T. In situ Mn K-edge X-ray absorption spectroscopic studies of anodically deposited manganese oxide with relevance to supercapacitor applications. J. Power Sources 2007, 166, 590–594.
  23. Chen, H.; He, J.; Zhang, C.; He, H. Self-assembly of novel mesoporous manganese oxide nanostructures and their application in oxidative decomposition of formaldehyde. J. Phys. Chem. C 2007, 111, 18033–18038.
  24. ->[33] Zhao, Q.; Yan, Z.; Chen, C.; Chen, J. Spinels: controlled preparation, oxygen reduction/evolution reaction application, and beyond. Chem. Rev. 2017, 117, 10121–10211.
  25. Selampinar, F.; Toppare, L.; Akbulut, U.; Yalçin, T.; Süzer, Ş. A conducting composite of polypyrrole II. As a gas sensor. Synth. Met. 1995, 68, 109–116.

47.-> [34] An, K.H.; Jeong, S.Y.; Hwang, H.R.; Lee, Y.H. Enhanced sensitivity of a gas sensor incorporating single‐walled carbon nanotube–polypyrrole nanocomposites. Adv. Mater. 2004, 16, 1005–1009.

  1. ->[35] Pei, Z.; Ma, X.; Ding, P.; Zhang, W.; Luo, Z.; Li, G. Study of a QCM dimethyl methylphosphonate sensor based on a ZnO-modified nanowire-structured manganese dioxide film. Sensors 2010, 10, 8275–8290.
  2. ->[36] Ramesh, S.; Lee, Y.-J.; Msolli, S.; Kim, J.-G.; Kim, H.S.; Kim, J.-H. Synthesis of a Co 3 O 4@ gold/MWCNT/polypyrrole hybrid composite for DMMP detection in chemical sensors. RSC Adv. 2017, 7, 50912–50919.
  3. ->[37] Sayago, I.; Matatagui, D.; Fernández, M.J.; Fontecha, J.L.; Jurewicz, I.; Garriga, R.; Muñoz, E. Graphene oxide as sensitive layer in Love-wave surface acoustic wave sensors for the detection of chemical warfare agent simulants. Talanta 2016, 148, 393–400.

51.-> [38] Lavoie, J.; Srinivasan, S.; Nagarajan, R. Using cheminformatics to find simulants for chemical warfare agents. J. Hazard. Mater. 2011, 194, 85–91.

  1. ->[39] Sun, L.; Wang, L.; Tian, C.; Tan, T.; Xie, Y.; Shi, K.; Li, M.; Fu, H. Nitrogen-doped graphene with high nitrogen level via a one-step hydrothermal reaction of graphene oxide with urea for superior capacitive energy storage. Rsc Adv. 2012, 2, 4498–4506.
  2. Ramesh, S.; Yadav, H.M.; Shinde, S.K.; Bathula, C.; Lee, Y.-J.; Cheedarala, R.K.; Kim, H.-S.; Kim, H.S.; Kim, J.-H. Fabrication of nanostructured SnO2@ Co3O4/nitrogen doped graphene oxide composite for symmetric and asymmetric storage devices. J. Mater. Res. Technol. 2020, 9, 4183–4193.
  3. Lee, Y.-J.; Kim, G.-H.; Jung, D.; Kim, J.-H. Dimethyl Methylphosphonate Detection Using a Quartz Crystal Microbalance as Chemical Warfare Sensor. Nanosci. Nanotechnol. Lett. 2015, 7, 1015–1018.
  4. ->[40] Kim, E.; Kim, J.; Ha, S.; Song, C.; Kim, J.-H. Improved Performance of Surface Acoustic Wave Sensors by Plasma Treatments for Chemical Warfare Agents Monitoring. J. Nanosci. Nanotechnol. 2020, 20, 7145–7150.
  5. ->[41] Hersee, S.D.; Ballingall, J.M. The operation of metalorganic bubblers at reduced pressure. J. Vac. Sci. Technol. A Vacuum, Surfaces, Film. 1990, 8, 800–804.
  6. ->[42] Dean, J.A. Lange’s handbook of chemistry. Mater. Manuf. Process 1990, 5, 687–688.
  7. ->[43] Kim, Y.S.; Ha, S.C.; Yang, H.; Kim, Y.T. Gas sensor measurement system capable of sampling volatile organic compounds (VOCs) in wide concentration range. Sensors Actuators, B Chem. 2007, 122, 211–218.
  8. ->[44] Standford Research systems QCM200 Digital controller: Operation and Service Manual; Sunnyvale, CA, USA, 2011;
  9. ->[45] Curie, J.; Curie, P. An oscillating quartz crystal mass detector. Rendu 1880, 91, 294–297.
  10. ->[46] Sauerbrey, G. Verwendung von Schwingquarzen zur Wägung dünner Schichten und zur Mikrowägung. Zeitschrift für Phys. 1959, 155, 206–222.
  11. ->[47] Rayleigh, Lord On waves propagated along the plane surface of an elastic solid. Proc. London Math. Soc. 1885, 1, 4–11.

64.-> [48] White, R.M.; Voltmer, F.W. Direct piezoelectric coupling to surface elastic waves. Appl. Phys. Lett. 1965, 7, 314–316.

  1. ->[49] Benes, E.; Groschl, M.; Seifert, F.; Pohl, A. Comparison between BAW and SAW sensor principles. IEEE Trans. Ultrason. Ferroelectr. Freq. Control 1998, 45, 1314–1330.
  2. ->[50] Kim, J.; Park, H.; Kim, J.; Seo, B.-I.; Kim, J.-H. SAW Chemical Array Device Coated with Polymeric Sensing Materials for the Detection of Nerve Agents. Sensors 2020, 20, 7028.

66.-> [51] Ramesh, S.; Haldorai, Y.; Kim, H.S.; Kim, J.-H. A nanocrystalline Co 3 O 4@ polypyrrole/MWCNT hybrid nanocomposite for high performance electrochemical supercapacitors. RSC Adv. 2017, 7, 36833–36843.

  1. ->[52] Zheng, M.; Zhang, H.; Gong, X.; Xu, R.; Xiao, Y.; Dong, H.; Liu, X.; Liu, Y. A simple additive-free approach for the synthesis of uniform manganese monoxide nanorods with large specific surface area. Nanoscale Res. Lett. 2013, 8, 1–7.
  2. Chen, H.; He, J. Facile synthesis of monodisperse manganese oxide nanostructures and their application in water treatment. J. Phys. Chem. C 2008, 112, 17540–17545.
  3. ->[53] Šetka, M.; Drbohlavová, J.; Hubálek, J. Nanostructured polypyrrole-based ammonia and volatile organic compound sensors. Sensors 2017, 17, 562.
  4. ->[54] Ramesh, S.; Yadav, H.M.; Karuppasamy, K.; Vikraman, D.; Kim, H.-S.; Kim, J.-H.; Kim, H.S. Fabrication of manganese oxide@ nitrogen doped graphene oxide/polypyrrole (MnO2@ NGO/PPy) hybrid composite electrodes for energy storage devices. J. Mater. Res. Technol. 2019, 8, 4227–4238.
  5. ->[55] Huang, M.; Li, F.; Dong, F.; Zhang, Y.X.; Zhang, L.L. MnO 2-based nanostructures for high-performance supercapacitors. J. Mater. Chem. A 2015, 3, 21380–21423.
  6. ->[56] Bai, X.P.; Zhao, X.; Fan, W.L. Preparation and enhanced photocatalytic hydrogen-evolution activity of ZnGa 2 O 4/N-rGO heterostructures. RSC Adv. 2017, 7, 53145–53156.
  7. Ramesh, S.; Karuppasamy, K.; Kim, H.-S.; Kim, H.S.; Kim, J.-H. Hierarchical Flowerlike 3D nanostructure of Co 3 O 4@ MnO 2/N-doped Graphene oxide (NGO) hybrid composite for a high-performance supercapacitor. Sci. Rep. 2018, 8, 1–11.
  8. ->[57] Wahid, M.; Parte, G.; Phase, D.; Ogale, S. Yogurt: a novel precursor for heavily nitrogen doped supercapacitor carbon. J. Mater. Chem. A 2015, 3, 1208–1215.
  9. ->[58] Yoon, H.; Jang, J. Conducting‐polymer nanomaterials for high‐performance sensor applications: issues and challenges. Adv. Funct. Mater. 2009, 19, 1567–1576.
  10. Hatchett, D.W.; Josowicz, M. Composites of intrinsically conducting polymers as sensing nanomaterials. Chem. Rev. 2008, 108, 746–769.

77.-> [59] Geng, L.; Wu, S. Preparation, characterization and gas sensitivity of polypyrrole/γ-Fe2O3 hybrid materials. Mater. Res. Bull. 2013, 48, 4339–4343.

  1. Hamilton, S.; Hepher, M.J.; Sommerville, J. Polypyrrole materials for detection and discrimination of volatile organic compounds. Sensors Actuators B Chem. 2005, 107, 424–432.
  2. ->[60] Joulazadeh, M.; Navarchian, A.H. Ammonia detection of one-dimensional nano-structured polypyrrole/metal oxide nanocomposites sensors. Synth. Met. 2015, 210, 404–411.
  3. ->[61] Kwon, O.S.; Park, C.S.; Park, S.J.; Noh, S.; Kim, S.; Kong, H.J.; Bae, J.; Lee, C.-S.; Yoon, H. Carboxylic acid-functionalized conducting-polymer nanotubes as highly sensitive nerve-agent chemiresistors. Sci. Rep. 2016, 6, 1–7.
  4. ->[62] Gupta, V.K.; Yola, M.L.; Eren, T.; Atar, N. Selective QCM sensor based on atrazine imprinted polymer: its application to wastewater sample. Sensors Actuators B Chem. 2015, 218, 215–221.
  5. ->[63] Wang, Y.; Yang, M.; Liu, W.; Dong, L.; Chen, D.; Peng, C. Gas sensors based on assembled porous graphene multilayer frameworks for DMMP detection. J. Mater. Chem. C 2019, 7, 9248–9256.

83.-> [64] Chen, D.; Zhang, K.; Zhou, H.; Fan, G.; Wang, Y.; Li, G.; Hu, R. A wireless-electrodeless quartz crystal microbalance with dissipation DMMP sensor. Sensors Actuators B Chem. 2018, 261, 408–417.

  1. ->[65] He, W.; Liu, Z.; Du, X.; Jiang, Y.; Xiao, D. Analytical application of poly {methyl [3-(2-hydroxy-3, 4-difluoro) phenyl] propyl siloxane} as a QCM coating for DMMP detection. Talanta 2008, 76, 698–702.
  2. ->[66] Alizadeh, T.; Soltani, L.H. Reduced graphene oxide-based gas sensor array for pattern recognition of DMMP vapor. Sensors Actuators B Chem. 2016, 234, 361–370.
  3. ->[67] Lama, S.; Kim, J.; Ramesh, S.; Lee, Y.-J.; Kim, J.; Kim, J.-H. Highly Sensitive Hybrid Nanostructures for Dimethyl Methyl Phosphonate Detection. Micromachines 2021, 12, 648.
  4. ->[68] Haghighi, E.; Zeinali, S. Nanoporous MIL-101 (Cr) as a sensing layer coated on a quartz crystal microbalance (QCM) nanosensor to detect volatile organic compounds (VOCs). RSC Adv. 2019, 9, 24460–24470.
  5. ->[69] Segal, S.R.; Suib, S.L.; Tang, X.; Satyapal, S. Photoassisted decomposition of dimethyl methylphosphonate over amorphous manganese oxide catalysts. Chem. Mater. 1999, 11, 1687–1695.
  6. ->[70] Collins, G.E.; Buckley, L.J. Conductive polymer-coated fabrics for chemical sensing. Synth. Met. 1996, 78, 93–101.
  7. ->[71] Chevallier, E.; Scorsone, E.; Bergonzo, P. New sensitive coating based on modified diamond nanoparticles for chemical SAW sensors. Sensors Actuators B Chem. 2011, 154, 238–244.
  8. ->[72] Hu, N.; Wang, Y.; Chai, J.; Gao, R.; Yang, Z.; Kong, E.S.-W.; Zhang, Y. Gas sensor based on p-phenylenediamine reduced graphene oxide. Sensors Actuators B Chem. 2012, 163, 107–114.
  9. ->[73] Wang, Y.; Du, X.; Li, Y.; Long, Y.; Qiu, D.; Tai, H.; Tang, X.; Jiang, Y. A simple route to functionalize siloxane polymers for DMMP sensing. J. Appl. Polym. Sci. 2013, 130, 4516–4520.
  10. ->[74] Du, X.; Wang, Z.; Huang, J.; Tao, S.; Tang, X.; Jiang, Y. A new polysiloxane coating on QCM sensor for DMMP vapor detection. J. Mater. Sci. 2009, 44, 5872–5876.
  11. ->[75] Thomas, G.; Spitzer, D. 3D Core–Shell TiO2@ MnO2 Nanorod Arrays on Microcantilevers for Enhancing the Detection Sensitivity of Chemical Warfare Agents. ACS Appl. Mater. Interfaces 2021.
  12. ->[76] Bertilsson, L.; Potje-Kamloth, K.; Liess, H.-D.; Engquist, I.; Liedberg, B. Adsorption of dimethyl methylphosphonate on self-assembled alkanethiolate monolayers. J. Phys. Chem. B 1998, 102, 1260–1269.
  13. ->[77] Greenler, R.G. Infrared study of adsorbed molecules on metal surfaces by reflection techniques. J. Chem. Phys. 1966, 44, 310–315.
  14. Francis, S.; Ellison, A.H. Infrared spectra of monolayers on metal mirrors. JOSA 1959, 49, 131–138.
  15. ->[78] Bellamy, L.J. The IR-Spectra of Complex Molecules, vol. 205 1975.
